# Refined Tensorial Radiance Field: Harnessing Coordinate-Based Networks for Novel View Synthesis from Sparse Inputs

## Abstract

The multi-plane encoding approach has been highlighted for its ability to serve as static and dynamic neural radiance fields without sacrificing generality. This approach constructs related features through projection onto learnable planes and interpolating adjacent vertices. This mechanism allows the model to learn fine-grained details rapidly and achieves outstanding performance. However, it has limitations in representing the global context of the scene, such as object shapes and dynamic motion over times when available training poses are sparse. In this work, we propose refined tensorial radiance fields that harness coordinate-based networks known for strong bias toward low-frequency signals. The coordinate-based network is responsible for capturing global context, while the multi-plane network focuses on capturing fine-grained details. We demonstrate that using residual connections effectively preserves their inherent properties. Additionally, the proposed curriculum training scheme accelerates the disentanglement of these two features. We empirically show that the proposed method achieves comparable results to multi-plane encoding with high denoising penalties in static NeRFs. Meanwhile, it outperforms others for the task with dynamic NeRFs using sparse inputs. In particular, we prove that excessively increasing denoising regularization for multi-plane encoding effectively eliminates artifacts; however, it can lead to artificial details that appear authentic but are not present in the data. On the other hand, we note that the proposed method does not suffer from this issue.

## 1 Introduction

Neural Radiance Fields (NeRFs) have gained recognition for their ability to create realistic images from various viewpoints using the volume rendering technique (Mildenhall et al., 2021). Early studies have demonstrated that multi-layer perception (MLP) networks, combined with sinuosidal encoding, can effectively synthesize 3-dimensional novel views (Tancik et al., 2020; Sitzmann et al., 2020; Martin-Brualla et al., 2021; Barron et al., 2021; 2022). These studies have shown that simple coordinate-based MLP networks exhibit strong low-frequency bias, and incorporating wide-spectrum sinusoidal encoding allows for capturing both low and high-frequency signals. Subsequent works illustrated the importance of appropriate sinusoidal encoding in conjunction with target signals to enhance performance (Martel et al., 2021; Lindell et al., 2022; Shekarforoush et al., 2022). To expedite the learning process, approaches explicitly parameterizing spatial attributes through multi-plane combinations have been introduced (Chen et al., 2022; Chan et al., 2022). In contrast to the aforementioned approaches, these methods dramatically reduce training time and produce cleaner and more realistic images, albeit at the cost of greater memory requirements.

For broader real-world applicability, extensive efforts have focused on reliably constructing radiance fields in cases of sparse input data. After the emergence of dynamic scenes dealing with time sparsity, addressing data sparsity has gained more attention in this field, as NeRF models commonly face overfitting issues due to the lack of consistent data for 3 or 4-dimensional space (Pumarola et al., 2021). One set of solutions tackled this by leveraging a pretrained image encoder to compare rendered scenes against consistent 3D environments (Yu et al., 2021; Wang et al., 2021; Chen et al., 2021; Jain et al., 2021). Another approach incorporated additional information, such as depth or color constraints, to maintain 3-dimensional coherence (Deng et al., 2022; Yuan et al., 2022;

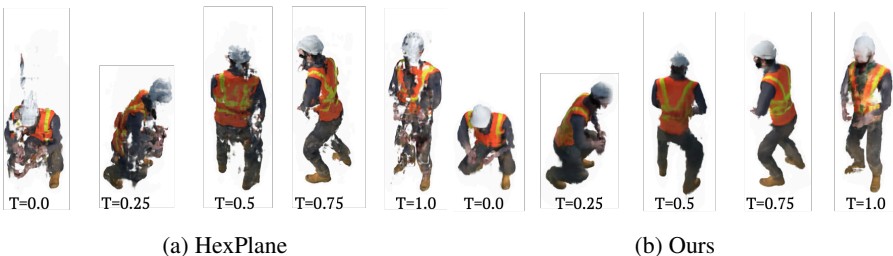

|  |  |  |  |  |  |  |  |  |  |
|---|---|---|---|---|---|---|---|---|---|
| T=0.0 | T=0.25 | T=0.5 | T=0.75 | T=1.0 | T=0.0 | T=0.25 | T=0.5 | T=0.75 | T=1.0 |

(a) HexPlane           (b) Ours

Figure 1: The qualitative results of the `standup` case in dynamic NeRFs using 25 training views (about 17% of the original data). This is challenging due to the limited information available along the time axis. Figure (a) is produced by HexPlane. (Cao & Johnson, 2023). Figure (b) is the rendered image of the proposed method.

Roessle et al., 2022; Truong et al., 2023). Methods progressively adjusting the frequency spectrum of position encoding have also proven effective in counteracting overfitting without additional information (Yang et al., 2023; Song et al., 2023).

However, a notable limitation of prior strategies dealing with sparse inputs is their less-than-ideal visual output. While the recent work reported successful reconstruction of static NeRF using voxel-grid parameterization in the sparse input regime with the assistance of denoising penalties like total variation (Sun et al., 2023), they often lack in adequately representing global elements like object morphology and dynamic motion, as evident in Figure 1a. Even if some renderings look crisp upon close inspection, the overall quality of the rendered results deteriorates due to the absence of global structures.

To alleviate this issue, we introduce a simple yet powerful approach to fundamentally improve the performance of static and dynamic NeRFs from sparse inputs. In this framework, the coordinate-based features are responsible for capturing global context, while the multiple-plane features are responsible for capturing fine-grained details. Moreover, in contexts with occlusions or time-variant dynamics, we employ a progressive weighting scheme that prevents the model from falling into local minima. This prioritizes low-frequency coordinate-based features to capture the global context first, allowing multiple-plane features to describe fine-grained target signals gradually. As a result, images generated by the proposed method exhibit improved clarity in terms of global contexts and fewer artifacts compared to baselines, as illustrated in Figure 1b. Our extensive experiments show that the proposed method achieves comparable results of multi-plane encoding with high denoising penalties in static NeRFs. Particularly, it outperforms baselines in dynamic NeRFs from the sparse inputs.

## 2 RELATED WORK

**Coordinate-based network and sinusoidal encoding** In the initial studies of NeRFs, MLP networks with sinusoidal encoding were used to simultaneously describe low and high-frequency details (Mildenhall et al., 2021; Martin-Brualla et al., 2021; Barron et al., 2021; 2022). However, it was found that a classical coordinate network without this encoding has a bias toward lower frequencies (Rahaman et al., 2019; Yüce et al., 2022). The importance of positioning encoding and sinusoidal activation led to the fundamental exploration of the relationship between rendering performance and the frequency values of target signals (Tancik et al., 2020; Sitzmann et al., 2020; Fathony et al., 2021; Ramasinghe et al., 2022). Lindell et al. (2022) uncovered that improper high-frequency embedding results in artifacts negatively impacting the quality of reconstruction. They addressed this issue using multi-scale bandwidth networks, where each MLP layer has a distinct spectrum of frequency embedding. Subsequent research utilized residual connections to faithfully maintain the designated spectrum without overwhelming high-frequency components (Shekarforoush et al., 2022).

**Explicit parameterization** Recent developments in explicit representations, such as voxel-grid, hash encoding, and multi-planes, have gained attention due to their fast training, rendering speed, and superior performance compared to positioning encoding-based networks (Liu et al., 2020; Sun et al., 2022; Müller et al., 2022; Chen et al., 2022; Cao & Johnson, 2023; Fridovich-Keil et al., 2023).

Sun et al. (2022) introduced the direct voxel field, using minimal MLP layers to speed up training and rendering. Instant-NGP, based on hash maps, provides multi-resolution spatial features and versatility, extending beyond 3-dimensional spaces to high-resolution 2-dimensional images (Müller et al., 2022). The multi-plane approach has been highlighted for its applicability in expanding to 4-dimensional without compromising generality, decomposing targets into multiple planes, with each plane responsible for a specific axis (Chen et al., 2022; Cao & Johnson, 2023; Fridovich-Keil et al., 2023). In particular, while the aforementioned approaches were executed on special on-demand GPU computations to boost efficiency, this method achieves comparable speed and performance based on general auto-differential frameworks. As a result, the multiple-plane approach broadens its scope to various tasks, including 3D object generation, video generation, 3D surface reconstruction, and dynamic NeRF (Gupta et al., 2023; Yu et al., 2023; Wang et al., 2023; Cao & Johnson, 2023; Fridovich-Keil et al., 2023).

**NeRFs in the sparse inputs** Early efforts incorporated pre-trained networks trained on large datasets to compensate for the lack of training data (Jain et al., 2021; Yu et al., 2021; Wang et al., 2021). Another alternative approach incorporated additional information, such as depth or color constraints, to ensure the preservation of 3D coherence (Deng et al., 2022; Yuan et al., 2022; Roessle et al., 2022; Truong et al., 2023). Without the assistance of off-the-shelf models and additional, this line of works devised new regularization to train NeRFs with fewer than ten views. Reg-NeRF incorporates patch-wise geometry and appearance regularization (Niemeyer et al., 2022). This paper verified their regularization performs well on forward-facing examples like DTU and LLFF datasets. They did not validate object-facing scenes because this assumption demands a high correlation between adjacent views. Recently, progressively manipulating the spectrum of positioning encoding from low to high-frequency proves effectiveness in mitigating over-fitting without relying on additional information (Yang et al., 2023; Song et al., 2023). Compared to explicit representations, those still suffer from unsatisfactory visual quality, characterized by blurry boundaries. Recent studies using total variation regularization on explicit representations get rid of artifacts and construct smoother surfaces (Cao & Johnson, 2023; Fridovich-Keil et al., 2023; Sun et al., 2023). However, our findings indicate that this regularization can introduce artificial details that seem real but are not in the data. This can also result in the model failing to converge in certain scenes. We present this problem in the experimental results, both qualitatively and quantitatively.

Another work attempted to use tri-planes with sinusoidal encoding of coordinates to create smoother surfaces (Wang et al., 2023), but their direction differs from our method since they mainly focus on enriching available features, as well as they did not demonstrate the role of tri-planes and coordinate features. In this paper, our new approach, refined tensorial radiance fields, proposes incorporating two distinct features: coordinate-based and multiple-plane features. We emphasize that the disentanglement of these two heterogeneous features is crucial for reliably constructing NeRFs in sparse inputs. The proposed method performs well even with higher-dimensional targets like dynamic NeRFs and extremely limited sparse inputs.

## 3 BACKGROUND

Before delving into the details of the proposed method, we briefly review the fundamentals of the neural radiance fields and multi-plane approach. We describe TensoRF (Chen et al., 2022) for the static NeRFs and HexPlane (Cao & Johnson, 2023) for the dynamic NeRFs. These methods are considered representative works in multi-plane encoding and are serve as main baselines in this paper.

### 3.1 NEURAL RADIANCE FIELDS

Mildenhall et al. (2021) proposed the original NeRF that uses volume rendering to compute predicted color values for novel view synthesis. In this framework, we consider a camera with origin $o$ and a ray direction $d$. A ray $\mathbf{r}$, composed of $n$ points, is constructed as $o + \tau_k \cdot d$, where $\tau_k \in \{\tau_1, \cdots, \tau_n\}$. The neural radiance field, parameterized by $\Theta$, predicts the color and density values $c_\Theta^k$, $\sigma_\Theta^k$ at each point. Using volume rendering, the predicted color value $\hat{\mathbf{c}}(\mathbf{r})$ are computed as follows; $\hat{\mathbf{c}}(\mathbf{r}; \Theta) = \sum_n T_n (1 - \exp(-\sigma_\Theta^k (\tau_{k+1} - \tau_k))) c_\Theta^k$. Here, the accumulated transmittance

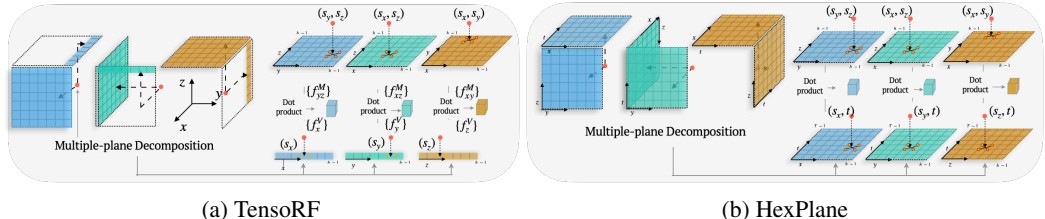

(a) TensoRF         (b) HexPlane

Figure 2: The schematic of baselines that use the multi-plane encoding. (a) TensoRF employs three planes and lines (Chen et al., 2022). (b) HexPlane adopts a total of six multiple planes to include the time axis (Cao & Johnson, 2023).

is computed by $T_n = \exp(-\sum_{k<n} \sigma_\Theta^k(\tau_{k+1} - \tau_k))$. The network parameters $\Theta$ are trained by minimizing the phometric loss, comparing $\hat{\mathbf{c}}(\mathbf{r})$ to the ground-truth color $\mathbf{c}$.

However, raw coordinate features alone are insufficient for describing high-frequency details. To resolve this, the paper proposes sinusoidal encoding, which transform coordinates into wide-spectrum frequency components. This encoding enables the description of both low and high-frequency signals, on the other hands, training can be time-consuming since it relies on implicit learning.

## 3.2 TENSORF: TENSORIAL RADIANCE FIELDS

The tensorial radiance fields provide an explicit parameterization using multiple-plane and fewer MLP layers. Compared to other explicit parameterization (Liu et al., 2020; Sun et al., 2022; Müller et al., 2022), multi-plane parameterization efficiently proves to be efficient for 3-dimensional NeRFs, provided that the plane resolution is sufficiently high. For simplicity, we assume that multi-planes share the same dimension in height, width, and depth denoted as $H$. This approach employs both plane features denoted as $\mathcal{M} = \{M_{xy}, M_{yz}, M_{zx}\}$ and vector features $\mathcal{V} = \{V_z, V_x, V_y\}$. For convenience, we denote two index variables, $i \in \{xy, yz, zs\}$ for $\mathcal{M}$ and $j \in \{z, x, y\}$ for $\mathcal{V}$. The plane and vector feature is denoted as $M_i \in \mathbb{R}^{c \times H \times H}$, $V_i \in \mathbb{R}^{c \times 1 \times H}$. Both plane and vector features have a channel dimensions $c$ to represent diverse information. To calculate the feature value at a given point $s := (s_x, s_y, s_z)$, the point are projected to corresponding planes and lines, and features on the nearest vertices are bilinear interpolated, as illustrated in Figure 2a. After obtaining the feature values from $\mathcal{M}$ and $\mathcal{V}$, denoted as $f^{\mathcal{M}} = \{f_{xy}^M, f_{yz}^M, f_{zx}^M\}$, and $f^{\mathcal{V}} = \{f_z^V, f_x^V, f_y^V\}$ and each feature $f_i \in \mathbb{R}^c$, hence $f^{\mathcal{M}}, f^{\mathcal{V}} \in \mathbb{R}^{3c}$. We use element-wise multiplication on $f^{\mathcal{M}}$, $f^{\mathcal{V}}$ to get final feature $f = f^{\mathcal{M}} \odot f^{\mathcal{V}} \in \mathbb{R}^{3c}$. For a more detailed explanation of multi-plane encoding, please refer to Appendix A. TensoRF has independent multi-plane features for density and appearance. TensoRF predicts occupancy by channel-wise summation of final density features across all planes. Conversely, appearance features are concatenated and then fed into MLP layers or spherical harmonics function.

Multiple-plane encoding is mainly designed to emphasize local representation with the nearest vertices. Therefore, TensoRF proposes gradually increasing the resolutions of the learnable planes and vectors during training to address this locality. This intends the model to learn the global context at the coarser resolution and then enhance finer details at the high resolution.

## 3.3 HEXPLANE

The following work, HexPlane, extends the multi-plane approach by incorporating the time axis, enabling it to work effectively in dynamic NeRFs. To achieve this, HexPlane builds upon the line features used in TensoRF, extending them into plane features by adding a time axis. This results in six planes, three spatial planes denoted as $\mathcal{M} = \{M_{xy}, M_{yz}, M_{zx}\}$, $M_i \in \mathbb{R}^{c \times H \times H}$ and three temporal planes $\mathcal{V} = \{V_{tz}, V_{tx}, V_{ty}\}$, $V_i \in \mathbb{R}^{c \times T \times H}$ as shown in Figure 2b. Likewise the previous subsection, we denote two index variables, $i \in \{xy, yz, zs\}$ for $\mathcal{M}$ and $j \in \{tz, tx, ty\}$ for $\mathcal{V}$. Compared to TensoRF, a key difference is that the sample $s := (s_x, s_y, s_z, t)$ includes the time variable. In dynamic NeRFs, dealing with temporal sparsity is a crucial factor for improving performance since the time axis contains relatively sparse information compared to spatial information.

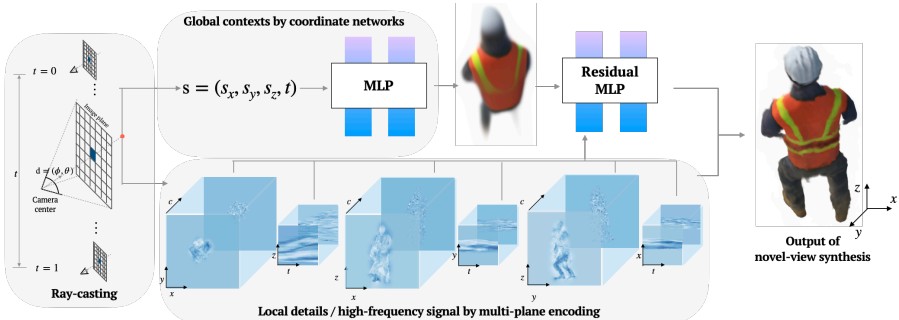

Figure 3: Conceptual illustration of the proposed method utilizing global contexts by coordinate networks and fine-grained details by multi-plane encoding. This method effectively displays two heterogeneous features. Notably, individual plane feature differs across channels, highlighting their disentanglement from other channels. All graphical representations are generated based on whether multi-plane features are masked or not, using our proposed method trained with 25 training views.

HexPlane addresses this challenge by employing denoising regularization, laplacian smoothing, that constrains similarity among adjacent multi-plane features. For an arbitrary plane feature $P$, Laplacian smoothing function $\mathcal{L}_l$ is defined as below, where $h, w$ refer row and column indices:

$$\mathcal{L}_l(P) = \sum_c \sum_{hw} \left( \left\| P^c_{h+1,w} - P^c_{h,w} \right\|^2_2 + \left\| P^c_{h,w+1} - P^c_{h,w} \right\|^2_2 \right). \tag{1}$$

Specifically, HexPlane applies laplacian smoothing on both plane features but give higher priority to temporal planes. This emphasize that time information is significant for capturing dynamic motion accurately. Fundamental operations of HexPlane align with TensoRF, including the direct prediction of density values by multi-plane features and the prediction of color values by concatenating multi-plane features, which are then fed into MLP layers.

## 4 REFINED TENSORIAL RADIANCE FIELDS: HARNESSING COORDINATE-BASED NETWORKS

We propose a novel method, referred to as "refined tensorial radiance field", that leverages coordinate-based networks. To mitigate the constraints of locality inherent in grid structures, our method capitalizes on a combination of distinct coordinate feature encoding techniques and multiplane representations, as depicted in Figure 3. subsection 4.1 illustrates the proposed residual-based architecture and the regularization strategy to facilitate the disentanglement of two heterogeneous features. In subsection 4.2, we explain a curriculum weighting strategy for multi-plane features. It ensures channel-wise disentanglement, providing a more diverse representation without the risk of overfitting where all channels exhibit identical expressions.

### 4.1 ARCHITECTURE AND LOSS FUNCTION

We describe how our model works in the dynamic NeRF case. Applying this model to a 3-dimensional static NeRF is feasible by simply excluding the $t$ variable. A key aspect of our network architecture is the utilization of coordinate-based networks along with explicit representation. In high-level context, we replace sinusodial encoding with multi-plane encoding while employing the architecture of the origianl NeRF. A coordinate $s := (s_x, s_y, s_z, t)$ is transformed via multiplane encoding from spatial and temporal plane features $\mathcal{M}, \mathcal{V}$ with element-wise multiplication $f = f^\mathcal{M} \odot f^\mathcal{V} \in \mathbb{R}^{3c}$. These features are then fed into MLPs parameterized by $\Theta$ along with their respective coordinates $s$. As shown by Shekarforoush et al. (2022), residual networks yield multi-fidelity results by preserving their pre-designated sinusoidal embeddings. In line with this, the proposed method adopts skip connections between acquired features and the hidden layer to serve the same purpose. Our empirical findings demonstrate that this operation promotes the disentanglement of two features, aligning with our intended purpose.

We introduce a loss function that combines photometric loss and laplacian smoothing across multiplane features. First, we define the photometric loss $\mathcal{L}_p$ as mean square errors between rendered

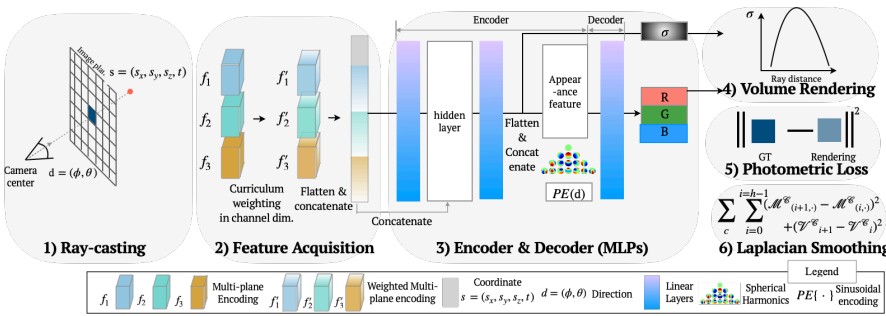

Figure 4: The schematic of the proposed method.

color $\hat{\mathbf{c}}(\mathbf{r})$ and ground truth pixel color $\mathbf{c}$, $\mathcal{L}_p(\Theta, \mathcal{M}, \mathcal{V}) = \sum_r \|\hat{\mathbf{c}}(\mathbf{r}; \Theta, \mathcal{M}, \mathcal{V}) - \mathbf{c}\|^2$. To tackle the ill-conditioned training problem in NeRFs arising from sparse-input situations, we apply Laplacian smoothing on both feature planes. Laplacian smoothing tends to excessively smooth signals, making them conform to global tendency rather than accurately local finer details (Sadhanala et al., 2017). Additionally, we regularize each plane feature using the L1 norm for the sparsity of multi-plane features. We use, $\|\mathcal{M}\|_1$ and $\|\mathcal{V}\|_1$ as $\sum_{i=1}^{i=3} \|M_i\|_1$ and $\sum_{i=1}^{i=3} \|V_i\|_1$ respectively. The entire loss function is as follows:

$$\mathcal{L}(\Theta, \mathcal{M}, \mathcal{V}) = \mathcal{L}_p(\Theta, \mathcal{M}, \mathcal{V}) + \lambda_1 \sum_{i=1}^{3} \left( \mathcal{L}_l(M_i) + \lambda_2 \mathcal{L}_l(V_i) \right) + \lambda_3 \left( \|\mathcal{M}\|_1 + \|\mathcal{V}\|_1 \right) \quad (2)$$

The only difference in the case of static NeRF comes from the dimension of $\mathcal{V}$. Laplacian loss is not applied to $\mathcal{V}$; the rest of the details are the same as in the 4D case. The hyperparameters and implementation detail can be found in Appendix B. While increasing the value of $\lambda_1$ allows to removes floating artifacts by over-smoothing the multi-plane features, it creates undesirable deformation that looks authentic but not be present in the training data. Hence, we opt not to utilize excessively high denoising weights. Instead, the coordinate network provides consistent training for multi-plane encoding when capturing high-frequency details. We empirically validate this through our experiments.

## 4.2 CURRICULUM WEIGHTING STRATEGY FOR MULTI-PLANE ENCODING

The architecture in the proposed method performs well in scenes with mild occlusion and less dynamic motion. However, it encounters challenges in severe ill-conditioned situations, such as heavy occlusion and rapid motion, as seen in the `drums` in the static NeRF and the `standup` in the dynamic NeRF. To alleviate this issue, we propose a curriculum weighting strategy for multi-plane encoding, aiming to manipulate the engagement of multi-plane features in accordance with training iterations. This approach trains the coordinate-based network first, followed by the subsequent training of multi-plane features. In this subsection, we denote $t$ as the training iteration. Technically, we introduce a weighting factor denoted as $\alpha(t)$ to control the degree of engagement of multi-plane features along the channel dimension of multi-planes. Here, $f = \{f_1, f_2, f_3\}$, and $f_i \in \mathbb{R}^c$ represents the output of multi-plane encoding, and the weighting factor $\gamma(t) = \{\gamma_1(t), \cdots, \gamma_c(t)\} \in \mathbb{R}^c$ is defined as follows:

$$\gamma_j(t) = \begin{cases} 0 & \text{if } \alpha(t) \le j \\ \frac{1 - \cos((\alpha(t) - j)\pi)}{2} & \text{if } 0 < \alpha(t) - j \le 1 \\ 1 & \text{otherwise,} \end{cases} \quad (3)$$

where, $j \in \{1, \cdots, c\}$ is the index of channel dimension and $\alpha(t) = c \cdot {(t-t_s)}/{(t_e-t_s)} \in [t_e, t_s]$ is proportional to the number of training iterations $t$ in the scheduling interval $[t_s, t_e]$. The final features $f'_i$ are obtained by $f'_i = f_i \odot \gamma(t)$. Hence, this weighting function is applied to each channel of multi-plane features. After reaching the last time-step of curriculum training, all channels of multi-plane features are fully engaged. It's worth noting that this weighting function is similar to those used in previous works such as (Park et al., 2021; Lin et al., 2021; Yang et al., 2023; Heo et al.,

Table 1: Result of evaluation statistics on the static NeRF datasets. We conduct five trials for each scene and report average scores. Average PSNR, SSIM, and LPIPS are calculated across all scenes. We indicates best performance as **bold** and second best as underline

| Models | PSNR ↑ | | | | | | | | Avg. PSNR ↑ | Avg. SSIM ↑ | Avg. LPIPS ↓ |
|---|---|---|---|---|---|---|---|---|---|---|---|
| | chair | drums | ficus | hotdog | lego | materials | mic | ship | | | |
| Simplified_NeRF | 20.35 | 14.19 | 21.63 | 22.57 | 12.45 | 18.98 | 24.95 | 18.65 | 19.22 | 0.827 | 0.265 |
| DietNeRF | 21.32 | 14.16 | 13.08 | 11.64 | 16.12 | 12.20 | 24.70 | 19.34 | 16.57 | 0.746 | 0.333 |
| HALO | 24.77 | 18.67 | 21.42 | 10.22 | 22.41 | 21.00 | 24.94 | 21.67 | 20.64 | 0.844 | 0.200 |
| FreeNeRF | 26.08 | 19.99 | 18.43 | 28.91 | 24.12 | 21.74 | 24.89 | 23.01 | 23.40 | 0.877 | 0.121 |
| DVGO | 22.35 | 16.54 | 19.03 | 24.73 | 20.85 | 18.50 | 24.37 | 18.17 | 20.57 | 0.829 | 0.145 |
| VGOS | 22.10 | 18.57 | 19.08 | 24.74 | 20.90 | 18.42 | 24.18 | 18.16 | 20.77 | 0.838 | 0.143 |
| iNGP | 24.76 | 14.56 | 20.68 | 24.11 | 22.22 | 15.16 | 26.19 | 17.29 | 20.62 | 0.828 | 0.184 |
| TensoRF | 26.23 | 15.94 | 21.37 | 28.47 | 26.28 | 20.22 | 26.39 | 20.29 | 23.15 | 0.864 | 0.129 |
| K-Planes | 27.30 | **20.43** | **23.82** | 27.58 | 26.52 | 19.66 | **27.30** | 21.34 | 24.24 | **0.897** | **0.085** |
| Ours | **28.02** | 19.55 | 20.30 | **29.25** | **26.73** | 21.93 | 26.42 | **24.27** | **24.56** | 0.896 | 0.092 |

2023). However, the key difference is a channel-wise weighting function for multi-plane features. This approach allows the decoding network to receive encodings from all channels of multi-plane features, with later-order channels being updated more slowly than earlier-order channels. Through our experiments, we found that this strategy effectively prevents all channels of multi-plane features from converging to similar patterns, thereby mitigating overfitting issues.

# 5 EXPERIMENTS

In this section, we present our experiments designed to address three pivotal questions: *1) Do existing sinusoidal embedding techniques effectively render clear scenes when given sparse input data? 2) Does the introduction of denoising regularizations enable explicit parameterization methods to consistently capture 3D coherence without artifacts with sparse input data? 3) Does the integration of disparate features, such as multiple planes and coordinates, substantially improve the performance of both static and dynamic NeRF?*

To answer those questions, we conducted vast experiments over scenarios of two sparse input cases: a few-shot static case and a 4-dimensional dynamic case. We also include ablation studies to substantiate the rationale behind the architectural choices in our proposed model. The design efficacy of our model is validated in two key areas: the reliance on regularization mechanisms and feature disentanglement.

We choose the datasets as in-ward-facing object poses, as they are more likely to be occluded by the objects from various viewing locations compared to forward-facing poses. For performance evaluation, we employ the PSNR metric to gauge the quality of image reconstruction. In addition, SSIM and LPIPS scores are reported to assess the perceptual quality of the rendered images. Further experimental details are described in Appendix C.

## 5.1 3-DIMENSIONAL STATIC RADIANCE FIELDS

We conducted 3-dimensional static NeRF experiments on the NeRF-synthetic dataset to evaluate whether our model adequately captures both the global context of a scene and fine details without introducing undesirable artifacts under sparse input conditions. Consistent with prior studies such as (Jain et al., 2021; Yang et al., 2023), we trained all models with 8 views. We compare our proposed models with sinusoidal encoding methods; Simplified NeRF, DietNeRF (Jain et al., 2021), HALO (Song et al., 2023) and FreeNeRF (Yang et al., 2023) and for explicit spatial parameterization methods; DVGO (Sun et al., 2022), VGOS (Sun et al., 2023), iNGP (Müller et al., 2022), TensoRF (Chen et al., 2022) and K-Planes (Fridovich-Keil et al., 2023). For all considered baselines, we applied regularization techniques that are congruent with their inherent characteristics and configurations.

The quantitative rendering results are shown in Table 1 and Figure 5. First, we observed that the proposed method outperforms the previous state-of-the-art method, FreeNeRF, in terms of both PSNR and perceptual quality. Sinusoidal encoding-based networks fail to capture high-frequency details and are prone to underfit in data with high-resolution structures, (ficus, lego). In con-

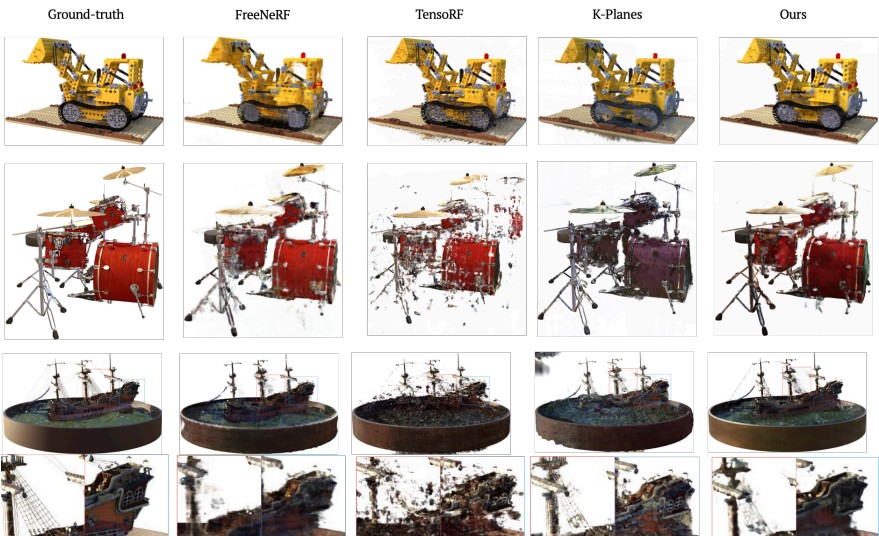

Figure 5: Rendered images of `lego`, `drums` and `ship` cases in the static NeRF dataset by FreeNeRF, TensoRF, K-Planes and ours. The rendered images are $\{83, 129, 95\}$-th in the test-set respectively.

trast, grid-based models show robust results in reconstructing high-frequency structures. However, for data with a strong non-Lambertian effect (`drums`, `ship`), grid-based models tend to miss the global shape and are prone to overfit in high-frequency. Our proposed multi-plane encoding technique can exclusively capture fine-grained details while maintaining global shape learned by coordinate features, leading to more robust novel view synthesis in sparse-input scenarios.

## 5.2 4-DIMENSIONAL DYNAMIC RADIANCE FIELDS

To demonstrate the robustness of the proposed model on more spare input cases, we conduct our experiences on the dynamic scenarios. We conducted 4-dimensional dynamic NeRF experiences on a D-NeRF data set. This data set comprises monocular cameras of about 50-100 frames duration and different in-ward facing views for each timestep. To verify a harsh situation, we also experimented with fewer frames $\{15, 20, 25\}$ sparse in both views and time aspects. Each view was sampled uniformly for each scene. To demonstrate the need for our refined tensorial radiance fields, we compare our method with HexPlane (Cao & Johnson, 2023) and its variants.

The observations made in subsection 5.1 are even more evident in the dynamic NeRFs. The proposed method outperforms every setting of HexPlane in all metrics in the D-NeRFs, as shown in Table 2. HexPlane discretizes the continuous time axis into finite bins, making it less responsive to the time-variant motion of objects when the available training poses are sparse. In contrast, the proposed method can capture the time-variant motion of objects by harnessing the coordinate-based networks first, with multi-plane encoding supplementing the remaining details. For instance, the variants of HexPlane do not accurately depict the shape of the blue ball over time, whereas the proposed method successfully does, including the reflection of light on the green ball. In the case of the `jumpingjack` sequence, the proposed method exhibits fewer artifacts and maintains the boundary of the scene better compared to HexPlane.

## 5.3 ABLATION STUDY

We assess the role of Total variation regularization or Laplacian smoothing within TensoRF, Hex-Plane, and the proposed method. In this experiment, we incrementally increase the parameter $\lambda_1$ from 0.0001 to 1.0, multiplying by a factor of 10. Table 3 demonstrate that our proposed method outperforms all experiment scenarios in both static and dynamic NeRF, with the sole exception of when $\lambda_1 = 0.001$ in the static NeRF. A notable performance difference was observed in the dynamic NeRF, which presents greater challenges due to time sparsity compared to the static NeRF. In detail,

Table 2: Result of evaluation statistics on the D-NeRF datasets. HexPlane employs the weight of denoising regularization as $\lambda_1 = 0.01$ via grid-search. Average PSNR, SSIM, and LPIPS are calculated across all scenes. We indicates best performance as **bold** for each cases

| Training views | Models | PSNR ↑ | | | | | | | | Avg. PSNR ↑ | Avg. SSIM ↑ | Avg. LPIPS ↓ |
|---|---|---|---|---|---|---|---|---|---|---|---|---|
| | | bouncingballs | hellwarrior | hook | jumpingjacks | lego | mutant | standup | trex | | | |
| 15 views | HexPlane | 26.56 | 15.91 | **21.03** | 20.35 | **23.64** | **23.40** | 21.48 | 23.05 | 21.93 | 0.921 | 0.092 |
| | K-Planes | 24.10 | 15.88 | 19.59 | 20.97 | 23.55 | 22.21 | 20.63 | **25.08** | 21.50 | 0.922 | **0.086** |
| | Ours | **28.09** | **16.48** | 20.90 | **21.51** | 23.54 | 23.38 | **21.87** | 24.88 | **22.30** | **0.925** | 0.087 |
| 20 views | HexPlane | 28.45 | 16.85 | 22.30 | 20.87 | **23.73** | 25.02 | **23.73** | 24.45 | 23.18 | 0.929 | 0.082 |
| | K-Planes | 25.43 | 17.25 | 21.07 | 21.40 | 23.12 | 25.01 | 21.01 | 25.84 | 22.58 | 0.931 | **0.070** |
| | Ours | **31.15** | **17.99** | **22.67** | **22.58** | 23.49 | **25.86** | 23.55 | **26.04** | **23.93** | **0.935** | 0.072 |
| 25 views | HexPlane | 30.49 | 17.61 | 23.10 | 22.85 | 24.29 | 25.81 | 23.74 | 25.30 | 24.15 | 0.935 | 0.074 |
| | K-Planes | 28.29 | 9.18* | 22.01 | 22.49 | **24.33** | 26.02 | 22.77 | **26.37** | 22.68 | 0.929 | 0.107 |
| | Ours | **34.61** | **19.21** | **23.82** | **24.46** | 23.78 | **26.75** | **26.07** | 26.29 | **25.34** | **0.941** | **0.063** |
| Full views | HexPlane | 39.21 | 23.92 | 27.97 | 30.53 | 24.74 | 32.19 | **33.09** | 30.02 | 30.15 | 0.964 | 0.039 |
| | K-Planes | 39.76 | 24.57 | 28.10 | 31.07 | **25.13** | **32.42** | 32.99 | **30.25** | **30.54** | **0.967** | **0.033** |
| | Ours | **40.25** | **24.63** | **28.50** | **31.70** | 25.09 | 31.19 | 31.45 | 29.76 | 30.20 | 0.960 | 0.049 |

\* indicates the model does not converge

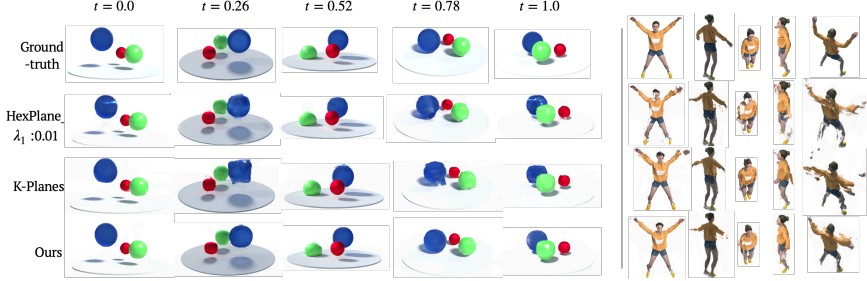

Figure 6: Rendered images of the `bouncingballs` and `jumpingjacks` in the dynamic NeRF dataset by HexPlane with $\lambda_1 = 0.01$, K-Planes and ours. All models are trained using 25 views

in the static NeRF dataset, our method yielded an average PSNR score between 22.99 and 24.55. In contrast, TensoRF with $\lambda_1 = 0.001$ performs the best at 24.98, but it fails to converge when $\lambda_1$ exceeds 0.01. This highlights that TensoRF is too sensitive and face challenges in training robustly with different regularization values. For the dynamic NeRF, HexPlane's scores ranged from 21.95 to 24.15, while ours spanned 24.67 to 25.74. This indicates our method is less dependent on denoising regularization, emphasizing the robust regularization capabilities of coordinate networks for multi-plane encoding. Our observations indicate that the proposed method maintains near-optimal performance across all scenarios once the $\lambda_1$ surpasses 0.001. This stability alleviates concerns about searching the regularization value for different scenes, significantly reducing hyperparameter tuning efforts. The detailed experimental results are included in Appendix E.

Furthermore, excessive regularization can introduce undesirable modification, including the introduction of color disturbances as evidenced in the case of `ship` with TensoRF, $\lambda_1 = 0.1$. Unlike the above, our method consistently achieves near-optimal performance without excessive denoising regularization, attributed to the coordinate-based networks capturing global contexts. As depicted in Figure 5, our method can restore fine geometries and reproduce accurate colors even under challenging conditions.

Table 3: Average PSNR across all scenes varying denoising regularization $\lambda_1$. The hyphen indicates not converged

| $\lambda_1$ | Static NeRF (8 views) | | | D-NeRF (25 views) | | |
|---|---|---|---|---|---|---|
| | TensoRF | K-Planes | ours | HexPlane | K-Planes | ours |
| 0.0001 | 24.10 | 24.31 | 23.68 | 22.83 | 24.32 | 24.67 |
| 0.001 | 24.98 | 24.28 | 24.47 | 23.86 | 24.01 | 25.38 |
| 0.01 | - | 24.28 | 24.55 | 24.15 | 24.02 | 25.74 |
| 0.1 | - | 23.64 | 24.23 | 23.46 | 23.55 | 25.84 |
| 1.0 | - | 22.05 | 22.99 | 21.95 | 22.62 | 25.42 |

# 6 CONCLUSION

In this paper, we introduce refined tensorial radiance fields that seamlessly incorporate coordinate networks. The coordinate network enables the capture of global context, such as object shapes in the static NeRF and dynamic motions in the dynamic NeRF dataset. This property allows multi-plane encoding to focus on describing the finest details.

ETHICS STATEMENT

Novel view synthesis is a task to understand the shape and appearance of objects and scenes from sparse set of images or video. Our model, in particular, can reconstruct fine detailed 3D shape with accurate appearance just from given fewer input both in static and dynamic scenes.

Like previous works, our model can obtain fine reconstruction results only if sufficiently distributed views are given. Recovering high fidelity 3D shape and appearance of objects from fewer inputs offers numerous practical applications. However, it also introduces potential drawbacks, such as the leading to the creation of potentially misleading media or potentially facilitating design theft, by duplicating physical objects.

REPRODUCIBILITY STATEMENT

Our code will be made publicly available upon publication. During the review process, we have attached our codes as supplementary files. For convenience reproducibility, both training and evaluation codes are included.

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

## A   MULTIPLE-PLANE ENCODING AND CONCATENATING COORDINATES

In this subsection, we discuss the use of multiple-plane encoding. Instead of directly predicting the density function using low-rank approximation of voxel grid, as done in previous methods (Chen et al., 2022; Cao & Johnson, 2023), our focus is on creating spatial features with multiple planes. For 3-dimensional data, we denote the plane features as $M_i \in \mathbb{R}^{c \times H \times H}$, and vector features $V_i \in \mathbb{R}^{c \times 1 \times H}$. However, in the case of 4-dimensional data, $V$ changes to plane

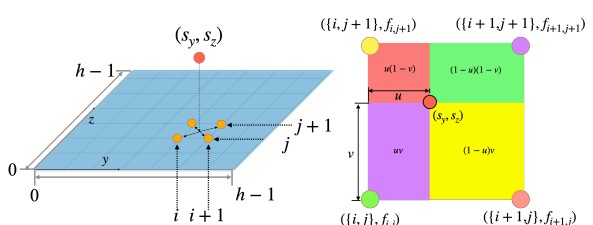

Figure A.1: Blinear interpolation

features. Each plane and vector feature corresponds to an axis in 3-dimensional spaces, such as $\mathcal{M} = \{M_{xy}, M_{yz}, M_{xz}\}$ and $\mathcal{V} = \{V_z, V_x, V_y\}$. In 4-dimensional spaces, the same notation applies to $\mathcal{M}$, but we introduce a time axis in $\mathcal{V}$ represented as $\mathcal{V} = \{V_{zt}, V_{xt}, V_{yt}\}$. The dimensions of $M_{(\cdot)}$ and $V_{(\cdot)}$ are $H \times W$ and $D$, respectively. We assume that all planes and vectors have the same dimension, i.e., $H = W = D$. We use $h$ as the all grid dimension for plane and vector features for simplicity.

To compute multiple-plane features, we use bilinear interpolation. In 3-dimensional data, when a data point $s \in \mathbb{R}^3$ is queried, it first drops to the axis for the corresponding dimension, then looks for the nearest vertices. For example, when obtaining plane features on $M_{x,y}$, $s = (s_x, s_y, s_z)$ drops $s_z$ and then looks for corresponding adjacent vertices in $M_1$. When $(i, j) = \lfloor (s_x, s_y) \rfloor$, the adjacent vertices are defined as $\{(i, j), (i + 1, j), (i, j + 1), (i + 1, j + 1)\}$, and their feature values are denoted as $\{f_{(i,j)}, f_{(i+1,j)}, f_{(i,j+1)}, f_{(i+1,j+1)}\}$ at the four nearest grid points. Here, $i, j \in \{0, 1, \cdots, h - 1\}$. The component of multiple-plane encoding $f(s_x, s_y)$ is computed by bilinear interpolation as follows:

$$f_{(s_x, s_y)} = (1 - u)(1 - v)f_{i,j} + u(1 - v)f_{i+1,j} + (1 - u)vf_{i,j+1} + uvf_{i+1,j+1} \qquad (A.1)$$

where, $u = (s_x - i)/(i+1-i)$ is the interpolation factor in the $x$-direction, and $v = (s_y - j)/(j+1-j)$ is the interpolation factor in the $y$-direction. The remaining components $(f_{(s_y, s_z)}, f_{(s_z, s_z)})$ are also computed by simply alternating coordinates. For the vector feature, we use linear interpolation, similar to bilinear interpolation but in 1 dimension. In 3-dimentional data, the features collected are $f^M = \{f_{(s_x, s_y)}, f_{(s_y, s_z)}, f_{(s_z, s_x)}\}$ and $f^V = \{f_{s_z}, f_{s_x}, f_{s_y}\}$, In 4-dimensional data, we can also use bilinear interpolation for $\mathcal{V}$. In this case, the features are $f^M = \{f_{(s_x, s_y)}, f_{(s_y, s_z)}, f_{(s_z, s_x)}\}$ and $f^V = \{f_{(s_z, t)}, f_{(s_x, t)}, f_{(s_y, t)}\}$. Then, we combine them by element-wise producting the two vectors $f = f^M \odot f^V$ to get multiple-plane encoding in $\mathbb{R}^{3c}$.

To represente low-frequencies signals apparently, we include the coordinate of a data point $s = \{s_x, s_y, s_z\} \in \mathbb{R}^3$ in 3-dimensional data. In 4-dimensinoal data, these coordinate features become $s = \{s_x, s_y, s_z, t\} \in \mathbb{R}^4$. The final result of encoding is the concatenation of two different features: $\mathbf{f} = \{f, s\}$. For 3-dimensional data, $\mathbf{f}$ is in $\mathbb{R}^{3+3c}$, and in case of 4-dimensional data, $\mathbf{f}$ is in $\mathbb{R}^{4+3c}$.

## B   IMPLEMENTATION DETAILS

### B.1   HYPER-PARAMETERS ON THE STATIC NERF

The proposed model consists of multi-plane encoding and MLPs with skip connections. For multi-plane encoding, we use 48 dimensional channels. The resolution of plane features is upsampled up to 8,000,000 ($200^3$) at the end of training. The weight for Laplacian smoothing($\lambda_1$), curriculum learning schedule, and initial feature resolution differ in each scene as we proceed with hyperparameter tuning for the optimal result. We listed detailed figures of hyperparameters of multi-plane encoding in Table B.1. For decoder MLP layers, we use standard fully connected layers with ReLU activations, 256 channels each. Our encoder contains four fully connected ReLU layers. We include

a skip connection after the second layers, which concatenates fused input features. The occupancy is calculated directly from the obtained features and is calculated as the `softplus` of the first channel. Then there is following RGB decoder consists of two layers. From the features obtained by RGB decoder, color values are obtained through `sigmoid` activation.

In our experiments, we trained over 30,000 iterations with batch size of 4,096. We use the Adam optimizer(Kingma & Ba, 2015) with initial learning rate is set to 0.02 and 0.001 for multi-plane features and for MLPs respectively, and followed learning rate scheduleling following TensorRF(Chen et al., 2022).

Table B.1: The detailed configuration for the static NeRF experiments. The parameters of curriculum $\{t_e, t_s\}$ are defined in Equation 3. These values are presented as a percentage of the total iteration. The hyphen means that curriculum learning does not apply.

| Configs | scenes | | | | | | | |
|---|---|---|---|---|---|---|---|---|
| | chair | drums | ficus | hotdog | lego | materials | mic | ship |
| $\lambda_1$ | 0.001 | 0.005 | 0.005 | 0.009 | 0.009 | 0.001 | 0.009 | 0.005 |
| curriculum learning | - | {5, 95} | - | - | {10, 50} | - | {0, 50} | - |
| Initial resolution | 16 | 3 | 3 | 24 | 48 | 48 | 48 | 3 |

## B.2 HYPER-PARAMETERS ON THE DYNAMIC NERF

The configuration for the dynamic NeRF case also follows the settings as in the static case. We utilize 48-channel plane features. The initial voxel resolution is set to 4,096 ($16^3$) and is subsequently upsampled to 8,000,000 ($200^3$). For more detailed descriptions, please refer to Table B.2. The structure of the decoder part, initial learning rate, and optimizer configuration remain identical to the static NeRF. Other configurations not specified here are taken directly from HexPlane's method as described in (Cao & Johnson, 2023).

Table B.2: The detailed configuration for the static NeRF experiments. The parameters of curriculum $\{t_e, t_s\}$ are defined in Equation 3. These values are presented as a percentage of the total iteration. The hyphen means that curriculum learning does not apply.

| Configs | scenes | | | | | | | |
|---|---|---|---|---|---|---|---|---|
| | boundingballs | hellwarrior | hook | jumpingjacks | lego | mutant | standup | trex |
| $\lambda_1$ | 0.001 | 0.005 | 0.001 | 0.001 | 0.05 | 0.001 | 0.05 | 0.05 |
| curriculum learning | - | {5, 95} | - | - | {5, 95} | - | {5, 95} | {5, 95} |

## C EXPERIMENTAL SETTING

We conducted the training and evaluation of all models using an NVIDIA A6000 with 48 GB of memory. For the experiments in Table 1, we utilized five different seeds: {0, 700, 19870929, 20220401, 20240507}. However, it's important to note that without explicitly describing the results for all five trials, each experiment was executed once, and the seed 0 was then used as the default. For explanations regarding the dataset and baselines, we provide the following description.

### C.1 DATASETS

**NeRF blender dataset** The Blender Dataset (Mildenhall et al., 2021) is a set of synthetic, bounded, 360°, in-ward facing multi-view images of static object. Blender Dataset includes eight different scenes. Following the previous method(Yang et al., 2023; Jain et al., 2021), for training, we used 8 views with IDs of 26, 86, 2, 55, 75, 93, 16, 73 and 8 counting from zeros. For evaluation, we uniformly sampled 25 images from the original test set. Unlike the evaluation settings in the previous works (Yang et al., 2023; Jain et al., 2021), we evaluate all metrics by using full-resolution images (800 × 800 pixels) for both training and testing. We downloaded Blender dataset from `https://www.matthewtancik.com/nerf`

**D-NeRF dataset**  D-NeRF Dataset (Pumarola et al., 2021) is a set of synthetic, bounded, 360 degree, monocular videos for dynamic objects. The D-NeRF dataset includes eight different scenes of varying duration, from 50 frames to 200 frames. To train the baseline under severe sparsity settings, we sub-sample the number of training views from the original D-NeRF dataset. For instance, in the case of `bouncingballs` that originally contains 150 views in the training set, we select a total of 25 views, evenly spaced apart, by starting from 0 and increasing by 6 at each step. For other scenes and varying number of views, we apply the same sampling method. We downloaded D-NeRF dataset from `https://github.com/albertpumarola/D-NeRF`

## C.2  BASELINES

In this chapter, we briefly explain the method we compared as a baseline in our experiments. About TensorRF and Hexplane we described in detail in section 3.

**Diet-NeRF**  Diet-NeRF (Jain et al., 2021) is a sinusoidal encoding based model. The model incorporates auxiliary semantic consistency loss which leverages pre-trained CLIP, networks trained on large data-sets to compensate for the lack of training data. Auxiliary semantic consistency loss regularize semantic similarity between rendered view and given input images. We also compare the simplified NeRF which is stated in Jain et al. (2021). For implementation we used the codebase in `https://github.com/ajayjain/DietNeRF`

**Free-NeRF**  Free-NeRF Yang et al. (2023) is a is a sinusoidal encoding based model. Yang et al. (2023) employed progressive activation of positioning embedding within a single model. It initially establishes global contextual shape and subsequentially describes fine-grained details. To reduce floating artifacts, it penalize near-camera density values, following the prior knowledge of object is located in a distance to the camera. For implementation we used the code from `https://github.com/Jiawei-Yang/FreeNeRF/tree/main`

**DVGO**  DVGO(Sun et al., 2022) is a model that uses a three-dimensional dense voxel feature grid. It utilizes independent voxel features for density and color. Shallow MLP follows color encoding. In the first stage, coarse geometry search learning, the initial shape is obtained, which provides the shape prior to the scene and finds empty voxels. Subsequently, in the fine reconstruction stage, they upsample the grid to a higher resolution and apply free-space skipping to optimize the occupied section densely. We used the code from `https://github.com/sunset1995/DirectVoxGO`

**Instant-NGP**  The Instant NGP(Müller et al., 2022) model expresses the voxel feature grid using the Hash function. It allocates features corresponding to each Voxel to the hash table, reducing the memory required while allowing collisions. Instant NGP utilizes the multi-resolution feature grid and uses features of resolution that log-scale uniformly increase from 16 to 1024-4096. It maintains a fast speed by inferring empty spaces through occlusion values such as TensorRF and DVGO and avoiding space sampling. We used the code from `https://github.com/kwea123/ngp_pl`

**VGOS**  VGOS(Sun et al., 2023) is the first example of applying the grid-based method to a Few-shot case. The method induces smoothness by adding total variation regularization to the dense grid feature, feature, depth, and color. In addition, progressive voxel sampling is introduced to prevent floating artifacts under the assumption that there will be a lot in the middle of the occlusion. We follows the code from `https://github.com/SJoJoK/VGOS`

**K-Planes**  K-planes utilizes the Hadamard product of multi-resolution tri-planes to represent voxel features. This approach extends from static three-dimensional scenes to dynamic four-dimensional NeRFs like Hex-plane(Cao & Johnson, 2023). K-Planes incorporates TV Loss and employs various regularization methods including distortion Loss(Barron et al., 2022) to reduce floating point artifacts. Furthermore, it adopts the proposal network method suggested in MipNeRF 360(Barron et al., 2022) as a sampling approach. We follow the code from `https://github.com/sarafridov/K-Planes`

# D THE RESULT STATISTICS OF 3-DIMENSIONAL STATIC NERF DATASET

From Table D.3 to Table D.5, we present the quantitative results for each scene of the synthetic NeRF Dataset. All reported numbers are averages of 5 experiments and corresponding standard deviations. Our model performed better in all metrics compared to all counterpart models. We also summarize the results of the TensorRF model with intense additional laplacian smoothness loss. The $\lambda_1 = 0.1$ value is the optimal value for obtaining the best value. This evidence is provided in Appendix E

TensoRF with strong Laplacian regularization shows comparable performance with the proposed model. The two methods show complement advantages in the novel-view rendering results. To compare, we present a novel-view renderings of `ship` (Figure 5) and `Mic`. TensorRF with $\lambda_1 = 0.1$ optimizes focusing on reconstruct higher-frequency texture. Therefore, it shows instability in low-frequency information such as a geometry (dec in `ship`), and show high-frequency artifacts on the color side (water regions in `ship`). Proposed method, TensoRefine has more strength in robust optimization, especially in global information. More accurate 3D geometry and view-consistent color reconstruction are possible. On the other hand, there are cases of underfitting in high resolution. Without relying on high denoising regularization, the proposed method nearly achieves the best performance, thanks to the coordinate-based networks responsible for capturing the global context.

Table D.3: The result of average PSNR in the static NeRF. We conduct five trials for each method and use 8 views for training.

| Models | PSNR ↑ | | | | | | | |
|---|---|---|---|---|---|---|---|---|
| | chair | drums | ficus | hotdog | lego | materials | mic | ship |
| Simplified_NeRF | 20.354 ±0.648 | 14.188 ±2.596 | 21.629 ±0.171 | 22.565 ±1.055 | 12.453 ±3.103 | 18.976 ±2.306 | 24.950 ±0.202 | 18.648 ±0.446 |
| DietNeRF | 21.323 ±2.478 | 14.156 ±5.143 | 13.082 ±3.892 | 11.644 ±6.753 | 16.120 ±7.121 | 12.200 ±7.343 | 24.701 ±1.222 | 19.342 ±4.033 |
| HALO | 24.765 ±0.280 | 18.674 ±0.226 | 21.424 ±0.204 | 10.220 ±0.388 | 22.407 ±1.997 | 20.996 ±0.032 | 24.937 ±0.078 | 21.665 ±0.229 |
| FreeNeRF | 26.079 ±0.545 | 19.992 ±0.050 | 18.427 ±2.819 | 28.911 ±0.232 | 24.121 ±0.633 | 21.738 ±0.085 | 24.890 ±1.733 | 23.011 ±0.148 |
| DVGO | 22.347 ±0.253 | 16.538 ±0.081 | 19.032 ±0.071 | 24.725 ±0.241 | 20.845 ±0.129 | 18.497 ±0.077 | 24.373 ±0.252 | 18.170 ±0.148 |
| VGOS | 22.100 ±0.036 | 18.568 ±0.112 | 19.084 ±0.061 | 24.736 ±0.073 | 20.895 ±0.073 | 18.418 ±0.036 | 24.180 ±0.148 | 18.155 ±0.060 |
| iNGP | 24.762 ±0.169 | 14.561 ±0.082 | 20.678 ±0.415 | 24.105 ±0.308 | 22.222 ±0.076 | 15.159 ±0.075 | 26.186 ±0.159 | 17.288 ±0.135 |
| TensoRF | 26.234 ±0.062 | 15.940 ±0.369 | 21.373 ±0.152 | 28.465 ±0.387 | 26.279 ±0.279 | 20.221 ±0.109 | 26.392 ±0.320 | 20.294 ±0.359 |
| TensorRF($\lambda_1 = 0.001$) | 28.527 ±0.208 | 19.626 ±0.134 | 21.963 ±0.217 | 29.373 ±0.218 | 29.441 ±0.270 | 21.911 ±0.087 | 26.998 ±0.325 | 22.837 ±0.717 |
| K-Planes | 27.300 ±0.192 | 20.427 ±0.153 | 23.820 ±0.215 | 27.576 ±0.254 | 26.520 ±0.262 | 19.661 ±0.178 | 27.297 ±0.144 | 21.337 ±0.240 |
| Ours | 28.021 ±0.143 | 19.550 ±0.587 | 20.301 ±0.258 | 29.247 ±0.656 | 26.725 ±0.565 | 21.927 ±0.114 | 26.416 ±0.199 | 24.266 ±0.163 |

Table D.4: The result of average SSIM in the static NeRF. We conduct five trials for each method and use 8 views for training.

| Models | SSIM ↑ | | | | | | | |
|---|---|---|---|---|---|---|---|---|
| | chair | drums | ficus | hotdog | lego | materials | mic | ship |
| Simplified_NeRF | 0.852 ±0.003 | 0.773 ±0.017 | 0.871 ±0.002 | 0.891 ±0.004 | 0.738 ±0.031 | 0.827 ±0.019 | 0.931 ±0.001 | 0.736 ±0.005 |
| DietNeRF | 0.857 ±0.025 | 0.716 ±0.133 | 0.653 ±0.123 | 0.705 ±0.111 | 0.709 ±0.148 | 0.662 ±0.166 | 0.933 ±0.011 | 0.731 ±0.043 |
| HALO | 0.883 ±0.001 | 0.822 ±0.003 | 0.877 ±0.002 | 0.806 ±0.064 | 0.827 ±0.032 | 0.847 ±0.003 | 0.931 ±0.000 | 0.763 ±0.001 |
| FreeNeRF | 0.908 ±0.003 | 0.852 ±0.001 | 0.866 ±0.008 | 0.942 ±0.002 | 0.871 ±0.003 | 0.862 ±0.001 | 0.935 ±0.010 | 0.778 ±0.003 |
| DVGO | 0.860 ±0.003 | 0.761 ±0.002 | 0.857 ±0.001 | 0.904 ±0.002 | 0.820 ±0.001 | 0.804 ±0.002 | 0.933 ±0.001 | 0.689 ±0.003 |
| VGOS | 0.857 ±0.001 | 0.834 ±0.001 | 0.859 ±0.000 | 0.905 ±0.000 | 0.824 ±0.000 | 0.804 ±0.001 | 0.932 ±0.001 | 0.686 ±0.001 |
| iNGP | 0.899 ±0.002 | 0.730 ±0.002 | 0.886 ±0.004 | 0.904 ±0.001 | 0.841 ±0.001 | 0.748 ±0.002 | 0.946 ±0.001 | 0.672 ±0.002 |
| TensoRF | 0.919 ±0.001 | 0.753 ±0.007 | 0.882 ±0.001 | 0.938 ±0.002 | 0.909 ±0.003 | 0.843 ±0.003 | 0.947 ±0.002 | 0.719 ±0.006 |
| TensorRF($\lambda_1 = 0.001$) | 0.943 ±0.001 | 0.856 ±0.001 | 0.901 ±0.001 | 0.945 ±0.001 | 0.941 ±0.002 | 0.873 ±0.001 | 0.955 ±0.002 | 0.772 ±0.006 |
| K-Planes | 0.935 ±0.001 | 0.869 ±0.002 | 0.925 ±0.001 | 0.949 ±0.001 | 0.921 ±0.002 | 0.850 ±0.001 | 0.958 ±0.001 | 0.767 ±0.003 |
| Ours | 0.931 ±0.001 | 0.860 ±0.011 | 0.881 ±0.002 | 0.948 ±0.003 | 0.914 ±0.005 | 0.879 ±0.001 | 0.949 ±0.001 | 0.802 ±0.002 |

Table D.5: The result of average LPIPS in the static NeRF. We conduct five trials for each method and use 8 views for training.

| Models | LPIPS ↓ | | | | | | | |
|---|---|---|---|---|---|---|---|---|
| | chair | drums | ficus | hotdog | lego | materials | mic | ship |
| Simplified_NeRF | 0.247 ±0.010 | 0.388 ±0.083 | 0.153 ±0.007 | 0.239 ±0.009 | 0.408 ±0.091 | 0.205 ±0.042 | 0.100 ±0.001 | 0.375 ±0.005 |
| DietNeRF | 0.177 ±0.051 | 0.382 ±0.253 | 0.447 ±0.201 | 0.539 ±0.225 | 0.339 ±0.254 | 0.426 ±0.282 | 0.079 ±0.021 | 0.278 ±0.069 |
| HALO | 0.134 ±0.003 | 0.234 ±0.012 | 0.109 ±0.012 | 0.417 ±0.113 | 0.149 ±0.066 | 0.167 ±0.012 | 0.098 ±0.004 | 0.290 ±0.007 |
| FreeNeRF | 0.101 ±0.005 | 0.142 ±0.003 | 0.138 ±0.068 | 0.069 ±0.001 | 0.092 ±0.003 | 0.107 ±0.002 | 0.094 ±0.029 | 0.228 ±0.003 |
| DVGO | 0.120 ±0.004 | 0.218 ±0.003 | 0.102 ±0.001 | 0.106 ±0.003 | 0.125 ±0.001 | 0.149 ±0.001 | 0.062 ±0.001 | 0.276 ±0.004 |
| VGOS | 0.124 ±0.001 | 0.201 ±0.002 | 0.100 ±0.001 | 0.104 ±0.001 | 0.123 ±0.000 | 0.148 ±0.001 | 0.063 ±0.001 | 0.278 ±0.001 |
| iNGP | 0.098 ±0.004 | 0.345 ±0.005 | 0.099 ±0.006 | 0.144 ±0.003 | 0.127 ±0.002 | 0.292 ±0.003 | 0.058 ±0.002 | 0.312 ±0.003 |
| TensoRF | 0.074 ±0.002 | 0.312 ±0.011 | 0.105 ±0.003 | 0.072 ±0.005 | 0.059 ±0.002 | 0.129 ±0.004 | 0.047 ±0.002 | 0.237 ±0.010 |
| TensorRF($\lambda_1 = 0.001$) | 0.047 ±0.001 | 0.132 ±0.009 | 0.066 ±0.001 | 0.050 ±0.001 | 0.037 ±0.002 | 0.069 ±0.001 | 0.037 ±0.001 | 0.186 ±0.007 |
| K-Planes | 0.052 ±0.002 | 0.107 ±0.005 | 0.061 ±0.002 | 0.054 ±0.001 | 0.051 ±0.002 | 0.116 ±0.003 | 0.036 ±0.001 | 0.199 ±0.005 |
| Ours | 0.078 ±0.001 | 0.139 ±0.022 | 0.082 ±0.003 | 0.064 ±0.005 | 0.057 ±0.005 | 0.067 ±0.002 | 0.059 ±0.001 | 0.191 ±0.004 |

# E EXPERIMENTS ON VARYING DENOISING WEIGHTS $\lambda_1$

We compare TensorRF for our model according to various degrees of denoising regularization.

Table E.6 shows the dependence of the TensorRF, and proposed model on denoising weight. According to the rendering results (figure), for TensoRF denoising does reduce floating artifacts, but, if it is too strong, undesired high-frequency artifacts appear, as described in Appendix D. This requires a searching process for regularization weight for the optimal value. On the other hand, our model is not dominantly affected denoising weights but shows better performance in various $\lambda_1$ values. In addition, even when the intensity of denoising increases, undesigned artifacts do not appear. This is possible because our model is a coordinate feature anchoring low-frequency information.

In the case of Dynamic, as sparsity increases, the result shows that denoising regularization alone is not enough for robust reconstruction. TensoRF added with smoothing shows degraded performance compared to the proposed model for all dynamic scenes. In addition, for TensoRF models, the optimal $\lambda_1$ value varies depending on the scene's characteristics, which explains that dependence on the smoothed loss becomes severe. In contrast, the proposed model consistently works well regardless of the weighting.

The above results show that our feature-fusion strategy already has sufficient robustness on sparse inputs. In addition, regularization shows synergy with our model design, as it assists in more realistic rendering without producing undesired artifacts, and it does works for extremely sparse input cases.

Table E.6: The comparison of Ours and TensoRF in the static NeRF dataset. We conduct experiments varying the value of $\lambda_1$. All models are trained using 8 views. We use seed 0 for reproducibility. The hyphen means that the model is not converged.

| Models | PSNR ↑ | | | | | | | | Avg. PSNR ↑ | Avg. SSIM ↑ | Avg. LPIPS ↓ |
|---|---|---|---|---|---|---|---|---|---|---|---|
| | chair | drums | ficus | hotdog | lego | materials | mic | ship | | | |
| TensoRF ($\lambda_1 = 0.0001$) | 27.15 | 16.85 | 21.84 | 29.35 | 28.03 | 21.41 | 26.99 | 21.17 | 24.10 | 0.880 | 0.103 |
| TensoRF ($\lambda_1 = 0.001$) | 28.24 | 19.94 | 21.94 | 29.46 | 29.04 | 22.03 | 26.62 | 22.58 | 24.98 | 0.898 | 0.078 |
| TensoRF ($\lambda_1 = 0.01$) | 27.97 | 20.04 | - | 29.22 | 28.93 | 21.98 | - | 23.24 | - | - | - |
| TensoRF ($\lambda_1 = 0.1$) | - | 19.80 | - | 28.12 | 27.11 | 21.37 | - | 21.93 | - | - | - |
| TensoRF ($\lambda_1 = 1.0$) | - | - | - | 25.97 | 24.55 | 19.36 | - | 22.24 | - | - | - |
| K-Planes ($\lambda_1 = 0.0001$) | 27.16 | 20.50 | 23.82 | 27.75 | 26.29 | 19.87 | 27.46 | 21.68 | 24.31 | 0.897 | 0.083 |
| K-Planes ($\lambda_1 = 0.001$) | 27.08 | 20.20 | 23.26 | 27.94 | 27.06 | 20.02 | 26.76 | 21.94 | 24.28 | 0.900 | 0.081 |
| K-Planes ($\lambda_1 = 0.01$) | 27.10 | 20.27 | 22.62 | 27.64 | 26.48 | 20.59 | 27.08 | 22.46 | 24.28 | 0.899 | 0.082 |
| K-Planes ($\lambda_1 = 1.0$) | 23.54 | 17.53 | 22.31 | 27.08 | 26.01 | 19.74 | 26.46 | 21.93 | 23.64 | 0.893 | 0.090 |
| K-Planes ($\lambda_1 = 0.1$) | 25.98 | 19.60 | 20.72 | 26.11 | 24.15 | 19.09 | 24.56 | 20.73 | 22.05 | 0.876 | 0.112 |
| Ours ($\lambda_1 = 0.0001$) | 27.79 | 17.67 | 19.30 | 28.62 | 24.81 | 21.49 | 26.16 | 23.57 | 23.68 | 0.884 | 0.111 |
| Ours ($\lambda_1 = 0.001$) | 27.94 | 19.04 | 20.07 | 29.13 | 27.26 | 21.85 | 26.93 | 23.55 | 24.47 | 0.893 | 0.091 |
| Ours ($\lambda_1 = 0.01$) | 27.61 | 19.21 | 20.17 | 29.51 | 27.31 | 21.55 | 26.74 | 24.27 | 24.55 | 0.895 | 0.098 |
| Ours ($\lambda_1 = 0.1$) | 27.07 | 19.60 | 20.55 | 29.09 | 25.43 | 22.50 | 26.13 | 23.56 | 24.23 | 0.889 | 0.108 |
| Ours ($\lambda_1 = 1.0$) | 25.12 | 17.99 | 19.89 | 27.64 | 22.74 | 21.98 | 25.55 | 23.05 | 22.99 | 0.876 | 0.136 |

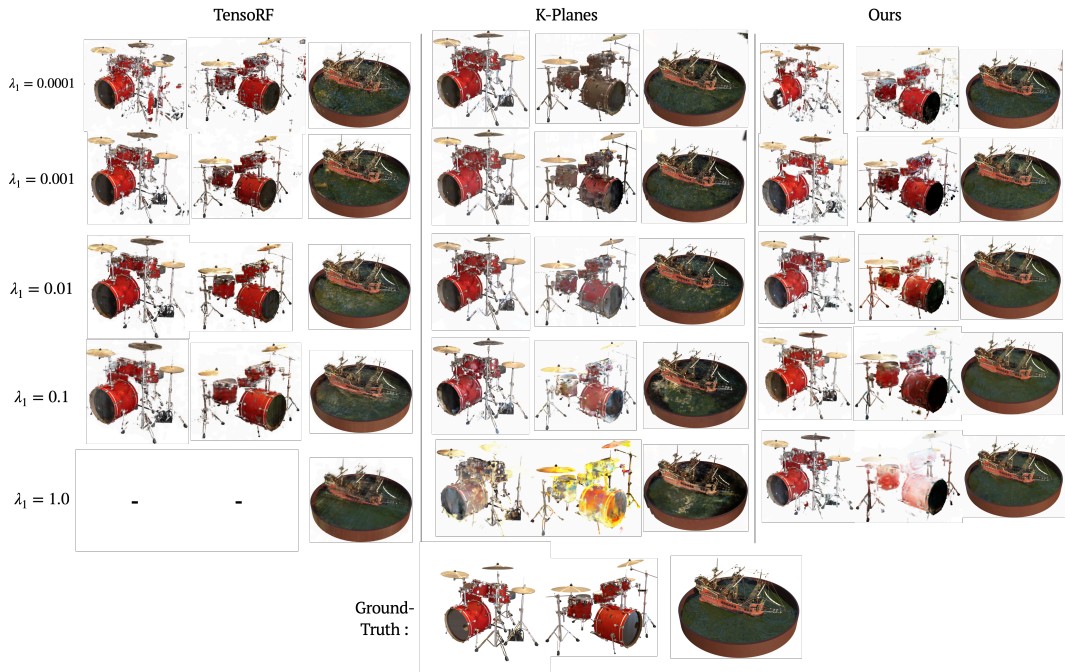

Figure E.2: Rendered images of `drums`, `ship` cases in the static NeRF dataset by TensoRF, K-Planes and ours with varying $\lambda_1$. To assess challenging scenarios, we select the 61st and 36th images for the `drums` scene and the 156th images for the `ship`.

Table E.7: The comparison of Ours and HexPlane in the dynamic NeRF dataset. We conduct experiments varying the value of $\lambda_1$. All models are trained using 25 views. We use seed 0 for reproducibility.

| Models | PSNR ↑ | | | | | | | | Avg. PSNR ↑ | Avg. SSIM ↑ | Avg. LPIPS ↓ |
|---|---|---|---|---|---|---|---|---|---|---|---|
| | bouncingballs | hellwarrior | hook | jumpingjacks | lego | mutant | standup | trex | | | |
| HexPlane ($\lambda_1 = 0.0001$) | 28.80 | 16.32 | 21.44 | 21.98 | 23.81 | 24.67 | 21.30 | 24.34 | 22.83 | 0.926 | 0.082 |
| HexPlane ($\lambda_1 = 0.001$) | 30.25 | 16.86 | 22.61 | 22.70 | 24.21 | 26.03 | 23.07 | 25.19 | 23.86 | 0.934 | 0.070 |
| HexPlane ($\lambda_1 = 0.01$) | 30.49 | 17.61 | 23.10 | 22.86 | 24.29 | 25.81 | 23.74 | 25.30 | 24.15 | 0.935 | 0.074 |
| HexPlane ($\lambda_1 = 0.1$) | 29.64 | 18.24 | 22.13 | 21.75 | 23.72 | 24.63 | 23.08 | 24.53 | 23.46 | 0.928 | 0.090 |
| HexPlane ($\lambda_1 = 1.0$) | 26.60 | 17.79 | 21.05 | 19.73 | 23.53 | 22.75 | 19.88 | 24.30 | 21.95 | 0.917 | 0.117 |
| K-Planes ($\lambda_1 = 0.0001$) | 29.39 | 16.72 | 22.69 | 23.98 | 24.03 | 26.42 | 24.47 | 26.88 | 24.32 | 0.937 | 0.074 |
| K-Planes ($\lambda_1 = 0.001$) | 29.22 | 17.92 | 22.29 | 22.73 | 24.12 | 26.20 | 23.22 | 26.35 | 24.01 | 0.939 | 0.061 |
| K-Planes ($\lambda_1 = 0.01$) | 29.38 | 18.29 | 22.33 | 22.78 | 23.82 | 26.18 | 23.02 | 26.33 | 24.02 | 0.938 | 0.062 |
| K-Planes ($\lambda_1 = 0.1$) | 28.85 | 17.53 | 21.52 | 22.52 | 24.02 | 26.00 | 22.74 | 25.25 | 23.55 | 0.931 | 0.074 |
| K-Planes ($\lambda_1 = 1.0$) | 25.29 | 17.90 | 20.99 | 21.63 | 23.61 | 25.06 | 21.73 | 24.73 | 22.62 | 0.928 | 0.087 |
| Ours ($\lambda_1 = 0.0001$) | 32.80 | 18.34 | 23.39 | 23.18 | 23.79 | 26.33 | 23.77 | 25.77 | 24.67 | 0.936 | 0.071 |
| Ours ($\lambda_1 = 0.001$) | 34.13 | 19.01 | 23.90 | 24.72 | 23.92 | 26.86 | 24.26 | 26.22 | 25.38 | 0.942 | 0.062 |
| Ours ($\lambda_1 = 0.01$) | 33.71 | 19.69 | 23.83 | 24.77 | 24.20 | 26.89 | 25.96 | 26.86 | 25.74 | 0.943 | 0.064 |
| Ours ($\lambda_1 = 0.1$) | 32.91 | 19.80 | 24.08 | 24.63 | 24.36 | 26.85 | 27.69 | 26.40 | 25.84 | 0.941 | 0.074 |
| Ours ($\lambda_1 = 1.0$) | 32.21 | 19.52 | 24.33 | 24.36 | 23.51 | 26.23 | 27.18 | 26.05 | 25.42 | 0.937 | 0.088 |

## F ABLATION STUDY ON ENCODING STRUCTURES

The encoder of our model applies the skip-connection of fused features. To justify design choice of our model, we compare the results of various encoder structures in static and dynamic cases Table E.6, Table F.9. All possible candidates regarding Encoder structures are listed and their graphical representations are also included in Figure F.4.

- Type 1 : Skip connection lies on every layer

- Type 2 : No skip connection, and employs fully connected MLPs

- Type 3 : Skip connection, but only coordinate $s$ is concatenated.

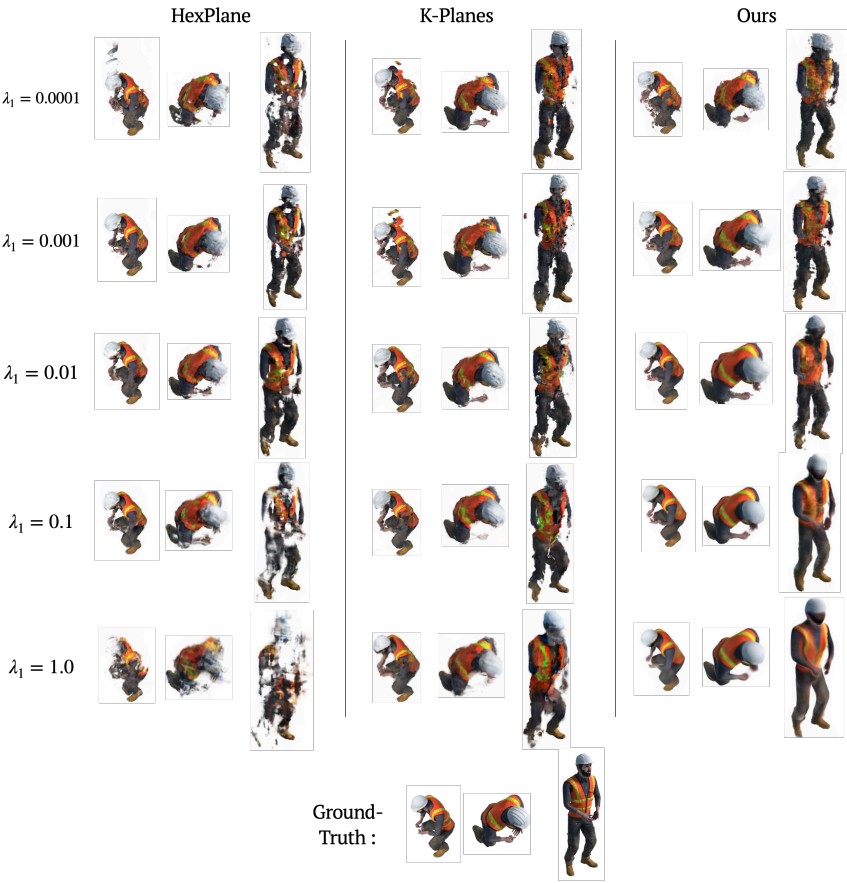

Figure E.3: Rendered images of standup cases in the dynamic NeRF dataset by HexPlane and ours with varying $\lambda_1$. We evaluate {0, 10, 19}th views in the test dataset.

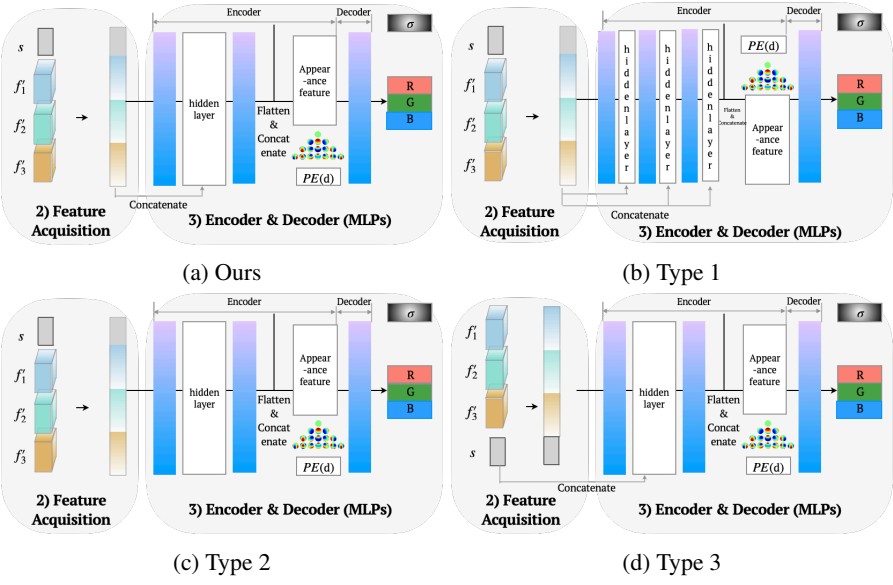

Figure F.4: The graphical representation for encoder structures used in Table F.8 and Table F.9.

Table F.8: The comparison of encoding structures. We evaluate four types of encoding structures including ours. All hyperparameters are consistent with those described in the original setting included Appendix B. All models are trained using 8 views in the static NeRF dataset. We use seed 0 for reproducibility.

| Models | PSNR ↑ | | | | | | | | Avg. PSNR ↑ | Avg. SSIM ↑ | Avg. LPIPS ↓ |
|---|---|---|---|---|---|---|---|---|---|---|---|
| | chair | drums | ficus | hotdog | lego | materials | mic | ship | | | |
| Ours | 28.15 | 20.09 | 20.04 | 29.43 | 27.58 | 22.06 | 26.41 | 24.18 | 24.74 | 0.898 | 0.089 |
| Type 1 | 23.83 | 17.85 | 19.14 | 18.45 | 20.54 | 12.97 | 14.61 | 22.78 | 18.77 | 0.844 | 0.179 |
| Type 2 | 26.15 | 18.02 | 19.53 | 17.78 | 19.73 | 11.72 | 18.06 | 22.87 | 19.23 | 0.848 | 0.171 |
| Type 3 | 25.16 | 19.40 | 19.33 | 17.94 | 20.88 | 11.85 | 14.62 | 23.35 | 19.07 | 0.843 | 0.175 |

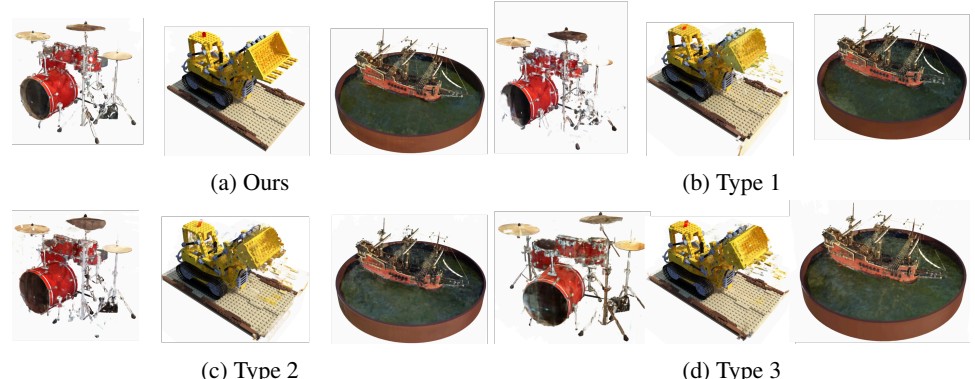

(a) Ours      (b) Type 1

(c) Type 2      (d) Type 3

Figure F.5: Rendered images are generated by alternating encoder structures. We selected the `drums`, `lego`, and `ship` scenes to follow the settings used in previous experiments.

Table F.9: The comparison of encoding structures. We evaluate four types of encoding structures including ours. All models are trained using 25 views in the dynamic NeRF dataset. All hyperparameters are consistent with those described in the original setting included Appendix B. We use seed 0 for reproducibility.

| Models | PSNR ↑ | | | | | | | | Avg. PSNR ↑ | Avg. SSIM ↑ | Avg. LPIPS ↓ |
|---|---|---|---|---|---|---|---|---|---|---|---|
| | bouncingballs | hellwarrior | hook | jumpingjacks | lego | mutant | standup | trex | | | |
| Ours | 33.83 | 18.93 | 23.54 | 24.24 | 23.69 | 26.59 | 26.06 | 26.05 | 25.37 | 0.942 | 0.063 |
| Type 1 | 33.99 | 18.01 | 24.01 | 24.26 | 23.91 | 26.95 | 24.55 | 26.56 | 25.28 | 0.941 | 0.064 |
| Type 2 | 33.35 | 18.08 | 23.82 | 24.58 | 24.08 | 26.85 | 24.46 | 26.84 | 25.26 | 0.941 | 0.063 |
| Type 3 | 32.74 | 18.64 | 24.24 | 24.83 | 23.99 | 27.08 | 25.17 | 26.81 | 25.44 | 0.942 | 0.062 |

In the case of the Dynamic case, smoothness induction in the temporal axis is essential, so the case of Type 3, using only the coordinate feature for skip-connection, shows slightly better performance than ours. Our model design works robustly, considering both static and dynamic cases, which verifies the suitability of model design choices.

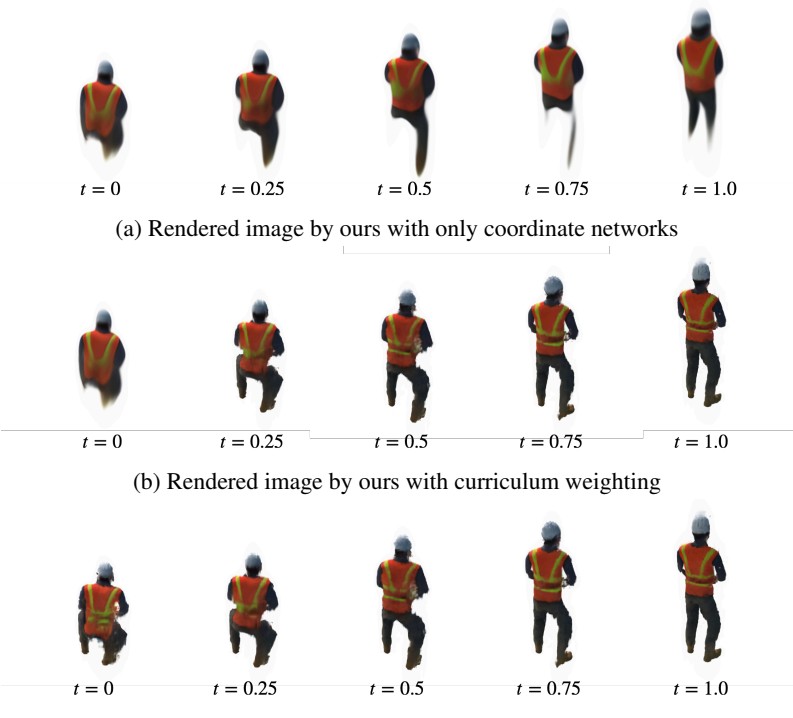

(a) Rendered image by ours with only coordinate networks

(b) Rendered image by ours with curriculum weighting

(c) Rendered image with full engangement of multi-plane

Figure G.6: Rendering results using different feature combinations. We show rendering results from three distinct combination of encoding features, (a) using only coordinates, (b) coordinates with progressively activating multi-plane encoding, and (c) full features. $t$ indicates the timesteps normalized to 1, and we use `standup` scene.

## G    DISENTANGLEMENT OF COORDINATE NETWORKS AND MULTI-PLANE ECODING

Our model learns by separating global shape and detail into coordinate and multi-plane features. Further, we adopt a progressive learning strategy on the channel axis among plane features to induce features to learn coarse-to-fine details. The proposed model shows robust reconstruction performance even when the input is highly sparse, as the proposed model successfully disentangle features into two aspects: (1) between heterogeneous features and (2) among channels in feature planes.

First, we analyze the disentanglement between heterogeneous features. We conducted the ablation analysis of the proposed method on the dynamic NeRFs with 25 training views. To qualitatively identify the role that coordinate-based networks in the proposed method, we separately evaluated the model using only the coordinate-based networks, with all multi-plane encodings set to zero. As shown in Figure G.6-(a), the coordinate-based networks are capturing the global context of the scene, including object shapes and large motions as we intended. It is worth noting that the coordinate-based networks can be effectively trained when used in conjunction with multi-plane encoding.

As previous studies have reported that high-frequency features in sinusoidal encoding tend to dominate, we anticipated that multi-plane encoding might overwhelm the coordinate-based networks (Lindell et al., 2022). However, the proposed architecture successfully disentangles and maintains these two heterogeneous features effectively throughout the training process. This demonstrates the synergy between the coordinate-based networks and multi-plane encoding in the proposed method.

Second, we analyze about channel-wise disentanglement among plane-features. To compare, we visualize multi-plane features of HexPlane and the proposed method, trained on full-view, and 25-views for `standup` scenes (Figure G.7). In `standup`, $z - x$ plane should encode the front shape of the person, and the $z - t$ plane should encode the upward movement.

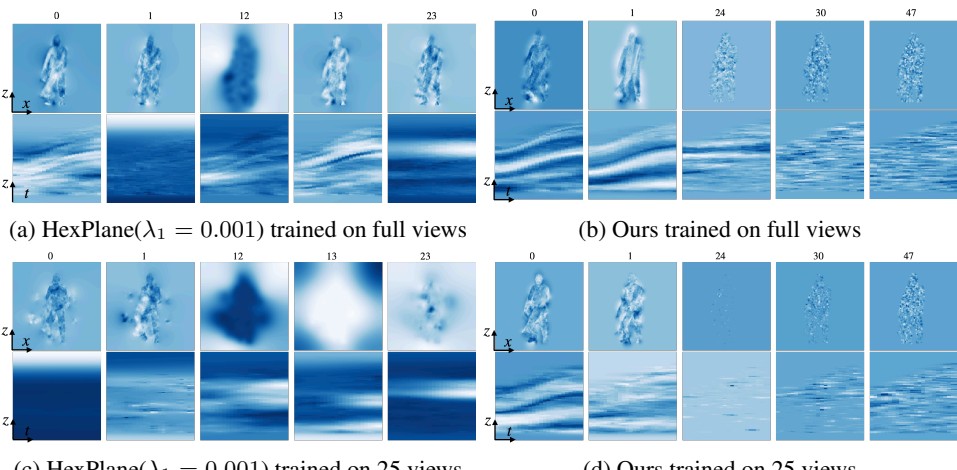

(a) HexPlane($\lambda_1 = 0.001$) trained on full views  (b) Ours trained on full views

(c) HexPlane($\lambda_1 = 0.001$) trained on 25 views  (d) Ours trained on 25 views

Figure G.7: Visualization of plane encoding features. We visualize 5 representative features from the plane encodings of Hexplane and Ours trained on `standup` scene.

For full-views, in HexPlane plane features (Figure G.7-(a)), global shape information and high-resolution detail are not distinguished and some channels even learn similar information which makes features less expressive. In contrast, the proposed method (Figure G.7-(b)) separates coarse-to-fine information along the channel axis, which effectively increases the expressiveness as each channel encodes information in different areas, In addition, our temporal features show a consistent upward tendency with distinct resolution features, while the hexplane learns similar resolution features with some wrong motions.

This trend is more clearly observable in fewer shots. In Figure G.7-(c), some spatial components result in overfitting or underfitting artifacts, and rightward information is hardly shown on the time axis. In contrast, our model (Figure G.7-(d)) maintains the coarse-to-fine manner. In particular, the trend remains the same with the time axis, confirming how much our progressive learning strategy has in a sparse setting.

From these observations, we verify that our two disentanglement strategies (inter-distinct features and inter-channel) are a way to learn global-to-detail features. This experiment allows it to analyze why the proposed model is more expressive and has robustness in sparse input.

## H    COMPARISON OF THE NUMBER OF PARAMETERS AND ANALYSIS OF TRAINING/RENDERING TIMES

Table H.10: We compare iNGP, TensoRF, K-Planes, and Ours on the static NeRF dataset by limiting the training steps to 15 million and 8 views used for training. The rendering time is assessed using 200 frames from the test dataset. Additionally, we conduct experiments to simplify the model parameters. The numbers in brackets represent the channel count in the multi-plane features. TensoRF(20) failed to train the scenes {chair, ficus, mic} due to training process instability, which we denote with a hyphen. In the K-Planes model, which features multi-resolutional multi-plane characteristics, we note that the total number of channel dimensions is the product of the number of resolutions and the channel dimension at each resolution.

| Model Name | # Params [M] | Avg. PSNR | Avg. Training Time [min] | Avg. Rendering Time [min] |
|---|---|---|---|---|
| iNGP (T=19) | 11.7M | 19.26 | 7.60 | 0.82 |
| iNGP (T=18) | 6.4M | 19.99 | 6.40 | 0.91 |
| K-Planes (3*16) | 17M | 23.95 | 17.61 | 6.83 |
| K-Planes (2*16) | 4.4M | 23.16 | 13.72 | 6.51 |
| TensoRF (64) | 17.3M | 25.23 | 7.72 | 7.82 |
| TensoRF (20) | 6.1M | - | - | - |
| Ours (48) | 6.0M | 24.36 | 31.16 | 46.02 |
| Ours (24) | 3.0M | 23.74 | 24.06 | 40.76 |

Table H.11: We compare TensoRF, K-Planes, and Ours on the dynamic NeRF dataset by limiting the training steps to 15 million and 25 views used for training. The rendering time is assessed using 20 frames from the test dataset. Additionally, we conduct experiments to simplify the model parameters. The numbers in brackets represent the channel count in the multi-plane features. In the K-Planes model, which features multi-resolutional multi-plane characteristics, we note that the total number of channel dimensions is the product of the number of resolutions and the channel dimension at each resolution.

| Model Name | # Params [M] | Avg. PSNR | Avg. Training Time [min] | Avg. Rendering Time [min] |
|---|---|---|---|---|
| K-Planes (3*32) | 18.6M | 23.85 | 18.93 | 0.83 |
| K-Planes (3*4) | 1.9M | 23.41 | 13.29 | 0.78 |
| HexPlane (72) | 9.7M | 24.00 | 6.78 | 0.60 |
| HexPlane (6) | 0.8M | 22.08 | 6.38 | 0.68 |
| Ours (48) | 3.4M | 25.17 | 12.22 | 2.14 |
| Ours (12) | 1.0M | 25.10 | 8.77 | 1.73 |

When there is insufficient training data, it is common practice to leverage the model's capacity and employ early stopping to achieve more robust and efficient learning. To investigate this trend, we conducted a comparison between TensoRF, K-Planes, and our method, limiting the training steps to 15 million and simplifying the model parameters. As shown in the Table H.10, the proposed method achieves comparable performance to TensoRF with a significantly smaller number of parameters. While TensoRF achieves optimal performance with 64 channels in multi-plane features, it shows instability in training and rendering times. Reducing its channels to 20 leads to convergence issues in specific scenes such as {chair, ficus, mic}, suggesting its limitations with sparse inputs.

In contrast, K-Planes is more stable than TensoRF, but it falls short in performance and requires more parameters, resulting in longer training and rendering times. On the other hand, the proposed method, despite not being the fastest in rendering compared to TensoRF and K-Planes, stands out for its stability in both training and rendering times, ensuring consistent performance. This stability is maintained even with fewer channels in the multi-plane features, making it more suitable for sparse inputs where robust training and stable performance are crucial. Moreover, we can further cut down rendering time by adopting the technique used in iNGP (Müller et al., 2022). This technique minimize redundant sampling by leveraging information from previous frames for predicting subsequent

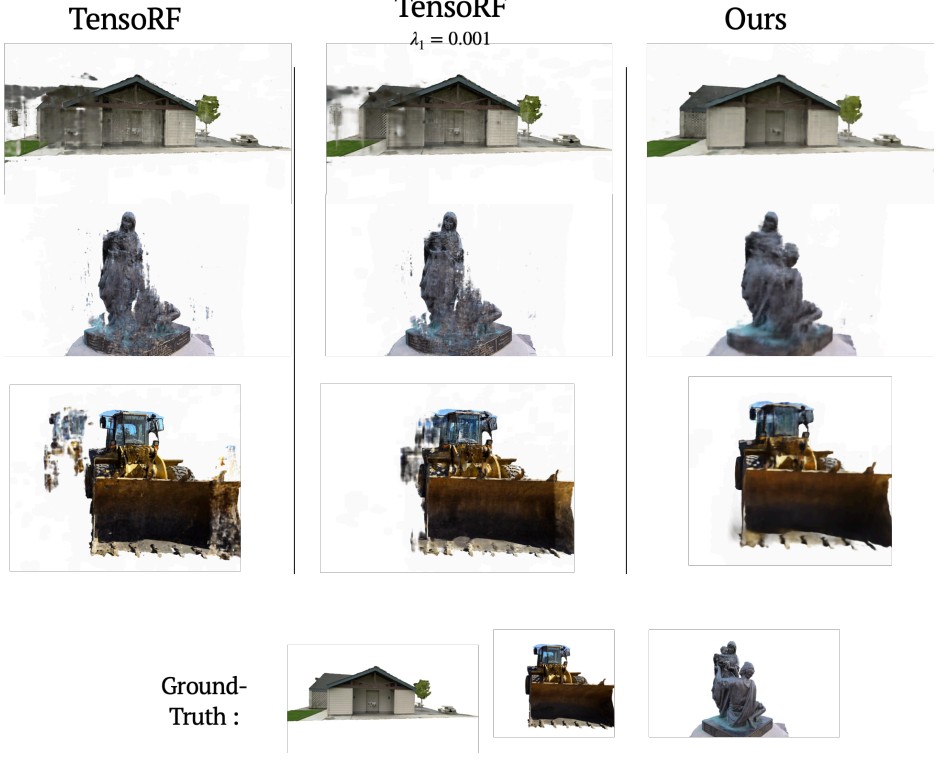

Figure I.8: The qualitative results of baselines and the proposed method on the Tanks and Temples dataset. We specifically show $\{47, 11, 12\}$-th images of `Barn`, `Family` and `Caterpillar` from the test dataset. We use $\{7, 10, 15\}$ percentiles of the training views for the `Caterpillar Barn` and `Family` scenes, respectively.

ones. This aspects underlines that in scenarios with sparse inputs, stability in training is more crucial than rendering speed.

Overall, our experiment underscores the strength of our method in maintaining stable training and rendering times, ensuring performance preservation, particularly in situations with sparse inputs where keeping a consistent performance across various hyper-parameters is important.

## I    EXPERIMENT ON REAL-WORLD DATASET : TANKS AND TEMPLES

The proposed method was evaluated on the real-world Tanks and Temples dataset (Knapitsch et al., 2017), where it was compared with the baseline TensoRF models, including a variation with a specific setting ($\lambda_1 = 0.001$). We focus on how each method handles the preservation of global context in scenes. As shown in Figure I.8, the proposed method consistently represent better rendered images than the baselines due to preserving the global context. This is a critical aspect when dealing with sprase input situations where maintaining the overall structure and shape of objects is essential. Despite TensoRF tends to focus on local details leading to partial but incomplete reconstructions seen in the case of `family`, our method excels in capturing the overall scene composition. This ability ensures that the larger structure and form of objects in the scene are accurately reconstructed, even at the cost of some finer details. Therefore, we demonstrate that the proficiency of our method becomes more apparent under conditions of sparse input data, and makes it particularly suitable for real-world applications where input data might be limited or incomplete.

Quantitatively, the proposed method shows its strength, especially in SSIM scores. While PSNR is a valuable metric for image quality, it can be biased in the context due to the lack of mask information and the inclusion of full-resolution white backgrounds. On the other hand, SSIM focuses on the

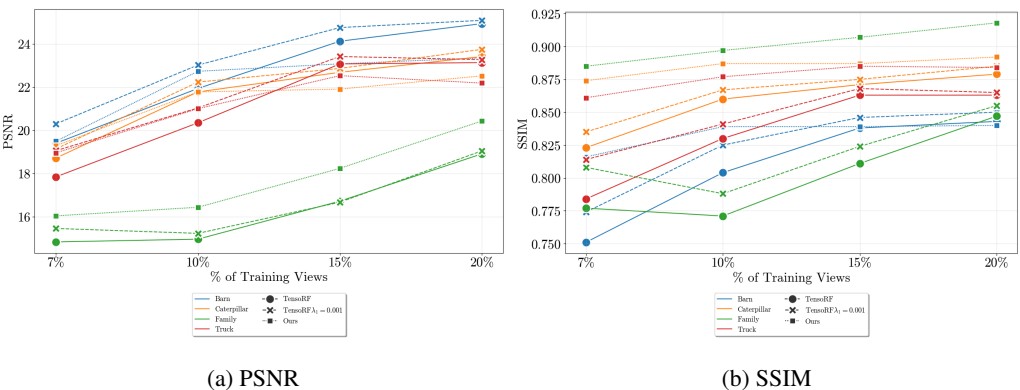

(a) PSNR

(b) SSIM

Figure I.9: The line plots of PSNR and SSIM on the Tanks and Temples dataset varying the number of training views.

perceived quality of structural information in the images. As shown in Figure I.9b, the proposed method consistently achieves higher SSIM scores across all scenes, indicating its superior capability in preserving the structural integrity and overall composition of scenes.

To sum up, the proposed method distinguishes itself from the baselines through its robust ability to preserve the global context of scenes, handle sparse input data effectively, and render images that are both structurally sound and visually realistic. These inherent properties highlight its potential for broader application in real-world scenarios, where input data is often sparse and incomplete.

