# OpenReview forum: "Refined Tensorial Radiance Field: Harnessing coordinate based networks for novel view synthesis from sparse inputs"
_ICLR.cc/2024/Conference — Submitted to ICLR 2024_

### Official Review · Reviewer_CJze · 2023-10-21

**Soundness:** 2 fair
**Presentation:** 3 good
**Contribution:** 2 fair
**Rating:** 6
**Confidence:** 5

**Summary:**

The paper proposes a method called refined tensorial radiance fields that combines coordinate-based networks and multi-plane encoding for novel view synthesis from sparse inputs. The propose method utilizes the propoties of coordinate-based networks to capture the low-freqency signals of the scene, and employs multi-plane encoding to focus on the high-frequency details. A curriculum training scheme is also proposed to progressively adjusts the weights of multi-plane features to prevent overfitting.
The paper conducts experiment on both static and dynamic NeRF datasets with sparse inputs. The proposed method outperforms the baselines in terms of PSNR, SSIM, and LPIPS metrics. Experiments also demonstrate the robustness and stability of the proposed method across different scenes and regularization values.

**Strengths:**

Strengths:
- Overall idea is simple and easy to understand. Writting is overall clear.
- Enough information is provided for reproduce. Code is also provided as supplemental materials to improve reproducibility.
- Dense ablation studies are provided in the paper and supplemental materials.

**Weaknesses:**

- This paper looks like a technical report more than a technical paper. The key insight I read from this paper is that "the coordinate-based features are responsible for capturing global context, while the multiple-plane features are responsible for capturing fine-grained details.". Yet there lacks further explanation and in-depth discussions related to this insight. Specifically, it is well known for NeRF that coordinate input itself struggles at fitting high-frequency details, and thus freq-based positional encoding and its variants [1, 2, 3] has been proposed to resolve this problem. Specifically, in [1] there are discussions and analysis (from NTK perspective) that coordinate input with Fourier representations is able to (1) learn high-freq details and (2) with some progressive learning strategy it is able to learning the frequency space in a coarse-to-fine manner. More technical (or theoretical) discussions regarding the coordinate input and its frequency behavior would make this paper far more interesting.

- Including coordinate inputs inevitably increase the computational cost, as each coordinate has to go through a (usually larger) MLP network instead of grid sampling and small MLP feed-forward as multi-plane feature input. In fact, one key motivation of multi-plane methods is decreasing the training and rendering time required. The paper seems not include any information and discussions regarding training time, computation cost, FPS, etc., which makes comparisons to multi-plane based methods incomplete.

[1] Tancik, Matthew, et al. "Fourier features let networks learn high frequency functions in low dimensional domains." Advances in Neural Information Processing Systems 33 (2020): 7537-7547.

[2] Barron, Jonathan T., et al. "Mip-nerf: A multiscale representation for anti-aliasing neural radiance fields." Proceedings of the IEEE/CVF International Conference on Computer Vision. 2021.

[3] Barron, Jonathan T., et al. "Mip-nerf 360: Unbounded anti-aliased neural radiance fields." Proceedings of the IEEE/CVF Conference on Computer Vision and Pattern Recognition. 2022.

**Questions:**

- Regarding the coordinate input: What kind of positional encoding is used?
- Regarding the results (especially for supp. video): How many input views used for objects shown in supp. video? It seems improvements over HexPlane on Dynamic NeRFs are subtle - there are still strong ghost artifacts between frames and the rendered image is noisy.

---

> ### Author Response · Authors · 2023-11-21
> **Response to Cjze's review (1/3)**
>
> Thank you for thoughtful comments.
>
> **Contribution of this paper** : This paper emphasizes that relying solely on explicit representation has limitations when dealing with sparse data. Interpolation-based plane features struggle to train reliably without an adequate amount of data points. Moreover, we found that a high level of denoising regularization can lead to color distortion, as evident in the case of K-Plane in Fig 5 and Fig E.2, and TensoRF often fails to converge in some instances. In contrast, our proposed method consistently represents the training datasets without artificial distortion. Even when excessive denoising regularization affects performance, as seen in the visual outputs in Fig 5 and Fig E.2, our approach avoids depicting artificial distortion, although the rendered images appear blurry. Therefore, we consistently demonstrate that the proposed method enhances performance and training stability through the integration of coordinate networks and multi-plane encoding. While our approach is straightforward and a combination of existing techniques, in this context, we believe it offers a valuable new axis for advancing NeRFs and improving their applicability.
>
> **Examining characteristics and analyzing explicit representation** : We conducted an ablation study to understand the role of each representation, and this content is included in Appendix F. In essence, our proposed method leverages the inherent properties of coordinate networks without sinusoidal encoding, which biases towards low-frequency details. During training, an increase in the engagement of multi-plane features does not lead to floating artifacts.
>
> The figures in Figure G.6 were not produced by separate trained models; they were all generated by our proposed method once it converged. After training, we masked all multi-plane features and named them coordinate networks. We found that coordinate networks are responsible for low-frequency details. As we increase the engagement of multi-plane features, the rendered images become finer and finer. This experiment solidly supports our claim that coordinate features handle low-frequency details while multi-plane features are responsible for finer details.
>
> **Theoretical aspects of the proposed method** :
>
> We acknowledge that theoretical aspects hold promise for future research directions. The significance of our experimental results is clear: existing explicit representations often overlook inappropriate embeddings in sparse regimes, and denoising regularization can introduce side effects like learning instability or color distortion.  Focusing on the core empirical findings and contributions can help ensure that the message is clear and impactful to target readers.
>
> We believe that the proposed approach can be linked to k-NN methods, which take into account global priors. A line of k-NN works has tackled hyper-parameter selection using adaptive methods by considering global patterns in relation to local details [1]. Similar to the original NeRF [2] and Fourier feature [3], we plan to explore this aspect further in the context of k-NN domains in future research.
>
> [1]Anava, Oren, and Kfir Levy. "k*-nearest neighbors: From global to local." Advances in neural information processing systems 29 (2016).
>
> [2]Mildenhall, Ben, et al. "Nerf: Representing scenes as neural radiance fields for view synthesis." Communications of the ACM 65.1 (2021): 99-106.
>
> [3]Tancik, Matthew, et al. "Fourier features let networks learn high frequency functions in low dimensional domains." Advances in Neural Information Processing Systems 33 (2020): 7537-7547.
>
> **Sinusodial encoding in the proposed method** :
>
> To clarify, the proposed method does not employ positioning encoding for coordinates. In the static NeRFs, the input is represented as $x \in \mathbb{R}^3$ for the coordinate network, while in the dynamic NeRFs, it is $x \in \mathbb{R}^4$. Positioning encoding is specifically used for ray direction. Additionally, for the case of ficus, we utilize positioning encoding for appearance features to avoid artificial color distortions. However, apart from this instance, the proposed method does not rely on positioning encoding for coordinates. The detailed information is described in Fig 4 and the attached configuration files.

---

> ### Author Response · Authors · 2023-11-21
> **Response to Cjze's review (2/3)**
>
> **The number of parameters and training time** : The proposed method does take longer compared to fast training methods like iNGP and TensoRF. However, it's worth noting that our approach still maintains reasonable training times. As shown in Table H.10, we've demonstrated that early stopping effectively preserves performance across all baselines, including our own, and all training procedures are completed around 30 minutes.
>
> We conduct a comparison between TensoRF, K-Planes, and our model on the static NeRF dataset, limiting the training steps to 0.15 million and 8 views used for training. Rendering time is evaluated using 200 frames from the test dataset. We also experiment with less parameterized models. The numbers in brackets indicate the channel count in the multi-plane features each axis. We note that TensoRF(20) encountered training instability issues and failed to train on the scenes {chair, ficus, mic}, denoted by a hyphen. In the K-Planes model, featuring multi-resolution multi-plane characteristics, the total number of channel dimensions is the product of the number of resolutions and the channel dimension at each resolution.
>
> |   Model  Name   | # Params [M] | Avg.  PSNR | Avg. Training  Time [min] | Avg. Rendering  Time [min] |
> |:---------------:|:------------:|:----------:|:-------------------------:|:--------------------------:|
> | K_Planes (3*16) |      17M     |    23.95   |           17.61           |            6.83            |
> | K_Planes (2*16) |     4.4M     |    23.16   |           13.72           |            6.51            |
> | TensoRF (64)    |     17.3M    |    25.23   |            7.72           |            7.82            |
> | TensoRF (20)    |     6.1M     |      -     |             -             |              -             |
> | Ours (48)       |     6.0M     |    24.36   |           31.16           |            46.02           |
> | Ours (24)       |     3.0M     |    23.74   |           24.06           |            40.76           |
>
> The following table shows the experiment result for dynamic NeRF, including the number of parameters, average training time and rendering time. We constrain the training steps to 0.15 million and 25 views used for training. Rendering time is evaluated using 20 frames from the test dataset.
>
> |   Model  Name   | # Params [M] | Avg.  PSNR | Avg. Training  Time [min] | Avg. Rendering  Time [min] |
> |:---------------:|:------------:|:----------:|:-------------------------:|:--------------------------:|
> | K_Planes (3*32) |      18.6M     |    23.85   |           18.93           |            0.83            |
> | K_Planes (3*4) |     1.9M     |    23.41   |           13.29           |            0.78            |
> | HexPlane (72)    |     9.7M    |    24.00   |            6.78           |            0.60           |
> | HexPlane (6)    |     0.8M     |    22.08       |           6.38            |            0.68             |
> | Ours (48)       |     3.4M     |    25.17   |           12.22           |            2.14           |
> | Ours (24)       |     1.0M     |    25.10   |           8.77           |            1.73           |
>   * We include the results of iNGP in the manuscript
>
>
> An intriguing point to note is that when we reduce the number of parameters in the baselines to match ours, such as K-Plane, and TensoRF, their performance suffers, and they sometimes fail to converge. However, when we halve the number of parameters, our model's reported performance only experiences a slight decrease while still outperforming the baselines, and it also achieves slightly faster training and rendering speeds.
>
> Regarding rendering time, our method may not be the most efficient. However, when combined with recent rendering techniques employed in iNGP, it can significantly accelerate rendering speeds. Nevertheless, we believe that this paper's primary focus is on robust training and criticizing the side-effects of existing regularization on explicit representation. Therefore, the rendering pipeline is not the main concern here, and this issue does not align with the core message of the paper.
>
> Regarding dynamic NeRFs, the training and rendering times do not exhibit a significant gap compared to static NeRFs. We will provide more details on this result shortly.

---

> ### Author Response · Authors · 2023-11-21
> **Response to Cjze's review (3/3)**
>
> **Qualitative results on dynamic NeRFs** : All the qualitative and visual results for dynamic NeRFs are based on the scenario with 25 views. In this analysis, even though the proposed method exhibits some foggy artifacts between frames, we note that when dealing with 25 views, learning time-dependent information becomes a challenging task in terms of training perspective. When compared to full views, we are only utilizing a maximum of 25% of the full dataset, and in terms of time information, the data provides coarser details about time. This creates an extremely challenging situation. Nevertheless, the proposed method does not produce floating artifacts or break the continuum structure, but also this model consistently tracks objects over time. As a result, the key point is that the proposed method remarkably enhances performance across all scenes, particularly for scenes like "bouncingballs" and "standup."

---

### Official Review · Reviewer_YmuZ · 2023-10-28

**Soundness:** 3 good
**Presentation:** 2 fair
**Contribution:** 2 fair
**Rating:** 5
**Confidence:** 3

**Summary:**

This paper proposes a simple but effective improvement for NeRF from sparse inputs, which combines multi-plane encoding with coordinate-based networks. Specifically, the coordinate-based network captures low-frequency structure and the multi-plane encoding is used to model the high-frequency details of the reconstruction. Experimental results justify the effectiveness of the proposed method.

**Strengths:**

- The proposed idea is easy to understand.
- The results are good and look visually-pleasing.

**Weaknesses:**

- The technical novelty is limited. Specifically, the proposed method is a simple combination of multi-plane encoding with coordinate-based networks, both of which are from previous methods, i.e., HexPlane and the original NeRF. The curriculum weighting strategy is more similar to a training trick. Few new technologies were proposed.
- It would be better to put the results side by side in the supplementary video to facilitate comparison.

**Questions:**

Please see weaknesses above.

---

> ### Author Response · Authors · 2023-11-21
> **Response to YmuZ's review**
>
> Thank you for your thoughtful comments.
>
> **Novelty and Contribution** : We've observed that when dealing with sparse inputs, relying solely on explicit representation tends to lead to overfitting the training data. While the regularization method mentioned by reviewer YmuZ does show some effectiveness, it's important to note that baseline models perform better under specific hyperparameters and scenarios. However, they often struggle to converge in other cases and can generate artifacts that appear like real data but are not part of the training set.
>
> In few-shot scenarios, the significance of stable training is widely recognized, even if the proposed techniques are not entirely fresh. We demonstrated that the previous methods still struggle to achieve consistent performance with sparse inputs as they tend to overly emphasize local details due to linear interpolation between feature grids rather than considering the low-frequency details. Even though we identify that classical approaches such as early-stopping and fewer capacity models are not effective in this setting, consequently, they cannot eliminate entirely all artifacts.
>
> More precisely, as shown in Table 1, numerical results reveal that K-Planes exhibit greater scene-to-scene variance compared to the proposed method. Qualitatively, as demonstrated in Fig 5 and Figure E.12, K-Planes exhibit color-related issues due to denoising regularization. Particularly, as the denoising regularization value increases, color distortion becomes more pronounced. TensoRF experiences instability problems when dealing with sparse inputs, as seen in Table 3 and Table E.6, failing to converge when $\lambda_1$ exceeds 0.01. Stability and consistent performance across different scenarios are crucial, especially in few-shot training settings, where achieving the best performance in a few cases may not generalize well to other experiments.
>
> On the other hand, Our proposed method consistently achieves top performance and robustly trains NeRFs, making it a reliable choice for various experimental cases. Furthermore, it outperforms other baselines in dynamic NeRF scenarios, indicating its stability and applicability in few-shot regimes. We have updated the literature survey to contain these arguments.
>
> **Additional supplementary animations** : We have uploaded as many supplementary videos as possible, including cases with varying denoising regularization.

---

### Official Review · Reviewer_D6bb · 2023-11-01

**Soundness:** 3 good
**Presentation:** 2 fair
**Contribution:** 2 fair
**Rating:** 3
**Confidence:** 4

**Summary:**

The paper proposes to use a hybrid of coordinates and multi-plane features for novel view synthesis from sparse inputs. The paper finds that coordinate-based network is better for capturing global context while the multi-plane features are better for fine-grained details. Therefore, the paper also proposes a curriculum training scheme for coarse-to-fine training. The coordinate network is first trained to learn global context, which provides a good initialization especially in the case of dynamic motion and sparse inputs. Then the multi-plane features are activated and increasingly weighted in accordance with the training iterations. Experiments show that the proposed method outperforms dynamic NeRFs using sparse inputs.

**Strengths:**

- The paper provides enough literature review and backgrounds.
- The paper proposes some effective techniques to improve the existing nerf model. The idea of using the hybrid of coordinate network (for global context) and multi-plane features (for details) seems reasonable to me, especially from the view of optimization. To better train the model, the author also provides a curriculum training scheme. The paper provides experiments to validate the idea.
- Codes are provided.

**Weaknesses:**

- The main concern for the paper is the limited novelty. Although the paper does provide some effective techniques, the majority part of the design heavily relies on several existing methods, such as the multi-plane features and the Laplacian smoothness. The combination of the existing two popular design (coordinate network and multi-plane features) seems direct and straight-forward to me. For me, the contribution of this paper is mainly technical (but still limited). It would be better if any theoretical explanation is provided.
- Some parts of the paper are not clear and some experiments might be needed (see Questions).

**Questions:**

- "The key difference is a channel-wise weighting function for multi-plane features": are there any quantitative ablation studies on the training scheme?
- Figure 3 is quite confusing, which is inconsistent with Figure 4. In Figure 3, is the coarse rendering image just for illustration or is it indeed generated as intermediate output?. What does the MLP and Residual MLP correspond to in Figure 4?
- Equation (3): the two conditions seem overlapped.
- What is the run time of the proposed method compared to baselines?

---

> ### Author Response · Authors · 2023-11-21
> **Response to D6bb's review (1/2)**
>
> Thanks for your thoughtful comments.
>
> **Contribution to NeRFs in the sparse inputs** :
> In few-shot scenarios, the significance of stable training is widely recognized, even if the proposed techniques are not entirely fresh. We demonstrated that the previous methods still struggle to achieve consistent performance with sparse inputs as they tend to overly emphasize local details due to linear interpolation between feature grids rather than considering the low-frequency details. Even though we identify that classical approaches such as early-stopping and fewer capacity models are ineffective in this setting, they cannot eliminate all artifacts.
>
> Our proposal suggests addressing both low-frequency and high-frequency details in sparse-input situations to avoid overfitting to only the finest details during training. In this context, our approach argues that a combination of multi-plane encoding and coordinate networks not only successfully captures both local and global patterns of the target signals but also provides a robust training method using channel-wise curriculum learning, emphasizing a global perspective before delving into finer details.
>
> **Theoretical aspects** : Focusing on the core empirical findings and contributions can help ensure that the message is clear and impactful to target readers. Therefore, it is entirely reasonable to consider excluding theoretical parts because they are out of the scope of this paper.
>
> While acknowledging that theoretical aspects hold promise for future research directions, our experimental results are crucial, so we want to emphasize our core message: existing explicit representations often overlook inappropriate embeddings in sparse regimes, and denoising regularization can introduce side effects like learning instability or color distortion.
>
> We believe that the proposed approach might be seen as akin to k-NN methods, taking into account global priors. A line of k-NN works have tackled hyper-parameter selection using adaptive methods [1]. Similar to the original NeRF [2] and Fourier feature [3], we plan to explore this aspect further in the context of k-NN domains in future research.
>
> [1]Anava, Oren, and Kfir Levy. "k*-nearest neighbors: From global to local." Advances in neural information processing systems 29 (2016).
>
> [2]Mildenhall, Ben, et al. "Nerf: Representing scenes as neural radiance fields for view synthesis." Communications of the ACM 65.1 (2021): 99-106.
>
> [3]Tancik, Matthew, et al. "Fourier features let networks learn high frequency functions in low dimensional domains." Advances in Neural Information Processing Systems 33 (2020): 7537-7547.
>
> **Explanation for Figure 3** : We agree with reviewer D6bb. The Figure 3 serves as a conceptual illustration of our method. The encoder mentioned in the Figure 4  encompasses the components of all layers in the Figure 3. Specifically, the 'MLP layer' refers to the linear layer preceding the hidden-layer in the Figure 4, while 'residual MLP' includes concatenation and the layers following the hidden layer. To clarify, the MLP denotes the coordinate networks, which are solely responsible for handling the coordinates. For example, $x \in \mathbb{R}^3$ in the case of static NeRFs, and $x \in \mathbb{R}^4$ in the case of dynamic NeRFs.
>
> However, we note that the images in Figure 3 were not produced by separate trained models; they were all generated by our proposed method once it converged. After training, we masked all multi-plane features and named them coordinate networks.

---

> ### Author Response · Authors · 2023-11-21
> **Response to D6bb's review (2/2)**
>
> **The number of parameters and training time** : The proposed method does take longer compared to fast training methods like iNGP and TensoRF. However, it's worth noting that our approach still maintains reasonable training times. As shown in Table H.10, we've demonstrated that early stopping effectively preserves performance across all baselines, including our own, and all training procedures are completed around 30 minutes.
>
> We conduct a comparison between TensoRF, K-Planes, and our model on the static NeRF dataset, limiting the training steps to 0.15 million and 8 views used for training. Rendering time is evaluated using 200 frames from the test dataset. We also experiment with less parameterized models. The numbers in brackets indicate the channel count in the multi-plane features each axis. We note that TensoRF(20) encountered training instability issues and failed to train on the scenes {chair, ficus, mic}, denoted by a hyphen. In the K-Planes model, featuring multi-resolution multi-plane characteristics, the total number of channel dimensions is the product of the number of resolutions and the channel dimension at each resolution.
>
> |   Model  Name   | # Params [M] | Avg.  PSNR | Avg. Training  Time [min] | Avg. Rendering  Time [min] |
> |:---------------:|:------------:|:----------:|:-------------------------:|:--------------------------:|
> | K_Planes (3*16) |      17M     |    23.95   |           17.61           |            6.83            |
> | K_Planes (2*16) |     4.4M     |    23.16   |           13.72           |            6.51            |
> | TensoRF (64)    |     17.3M    |    25.23   |            7.72           |            7.82            |
> | TensoRF (20)    |     6.1M     |      -     |             -             |              -             |
> | Ours (48)       |     6.0M     |    24.36   |           31.16           |            46.02           |
> | Ours (24)       |     3.0M     |    23.74   |           24.06           |            40.76           |
>
> The following table shows the experiment result for dynamic NeRF, including the number of parameters, average training time and rendering time. We constrain the training steps to 0.15 million and 25 views used for training. Rendering time is evaluated using 20 frames from the test dataset.
>
> |   Model  Name   | # Params [M] | Avg.  PSNR | Avg. Training  Time [min] | Avg. Rendering  Time [min] |
> |:---------------:|:------------:|:----------:|:-------------------------:|:--------------------------:|
> | K_Planes (3*32) |      18.6M     |    23.85   |           18.93           |            0.83            |
> | K_Planes (3*4) |     1.9M     |    23.41   |           13.29           |            0.78            |
> | HexPlane (72)    |     9.7M    |    24.00   |            6.78           |            0.60           |
> | HexPlane (6)    |     0.8M     |    22.08       |           6.38            |            0.68             |
> | Ours (48)       |     3.4M     |    25.17   |           12.22           |            2.14           |
> | Ours (24)       |     1.0M     |    25.10   |           8.77           |            1.73           |
>   * We include the results of iNGP in the manuscript
>
>
> An intriguing point to note is that when we reduce the number of parameters in the baselines to match ours, such as K-Plane, and TensoRF, their performance suffers, and they sometimes fail to converge. However, when we halve the number of parameters, our model's reported performance only experiences a slight decrease while still outperforming the baselines, and it also achieves slightly faster training and rendering speeds.
>
> Regarding rendering time, our method may not be the most efficient. However, when combined with recent rendering techniques employed in iNGP, it can significantly accelerate rendering speeds. Nevertheless, we believe that this paper's primary focus is on robust training and criticizing the side-effects of existing regularization on explicit representation. Therefore, the rendering pipeline is not the main concern here, and this issue does not align with the core message of the paper.
>
> Regarding dynamic NeRFs, the training and rendering times do not exhibit a significant gap compared to static NeRFs. We will provide more details on this result shortly.
>
> **Minor modification** : We revised the manuscript the Equation 3 to account for the disappearance of overlapped regions.

---

### Official Review · Reviewer_r4Vs · 2023-11-01

**Soundness:** 3 good
**Presentation:** 3 good
**Contribution:** 2 fair
**Rating:** 5
**Confidence:** 4

**Summary:**

The paper proposes a new approach for novel view synthesis from sparse input images with neural field representations. Based on  the existing work "TensorRF", this paper extends the representation with an additional coordinate-based network that should capture global context. Furthermore, the authors propose a progressive weighting scheme for enabling low-frequency modeling with the coordinate-based network and fine-grained details with a triplane representation.
The authors conduct experiments on static and dynamic scenes and compare them to various baseline methods such as TensoRF, INGP, FreeNeRF, HexPlane and more.

**Strengths:**

The paper is well written and the method is explained in detail.
The visuals in paper are very pleasing and help to understand the method, especially Figure 2 and 3.
Related work is well covered.
Various baseline methods are used.

**Weaknesses:**

1) In my view, the related work section misses concrete details on the relation of existing works to the proposed paper, e.g. in the “NeRFs in the sparse inputs” paragraph it stays unclear which of the mentioned limitations is resolved with the proposed method. Please provide more discussion here.

2) A major limitation of the experimental evaluation is that there is no discussion of training time and overall model size/ number of parameters. A major contribution of previous works (TensoRF and iNGP) is the extremely fast training time (TensoRF ~10min) and model compactness which mainly comes from the fact that only shallow MLPs are used. I’m wondering how the proposed approach performs in training time and model size.

3) It would be great to have experiments on more realistic data, e.g. for the dynamic case there is the  Plenoptic Video dataset used in the HexPlane paper.

4) The contribution is rather limited to extended architecture, additional loss function and a weighting strategy. Even though numbers improve, the technical novelty on top of existing work, e.g. Hexplanes and TensorRF is minor and it remains unclear if it is even worse in terms of training time and model size.

**Questions:**

It would be great to hear the authors opinion on the weaknesses 2) about the training time and model size. Can you provide numbers for the training time in comparison to the baselines on both tasks?

Please follow up on 1) to get a more concrete idea how the author's works position in this field.

**Details Of Ethics Concerns:**

-

---

> ### Author Response · Authors · 2023-11-21
> **Response to r4Vs's review (1/3)**
>
> Thank you for your thoughtful comments.
>
> **Contribution to NeRFs in the sparse inputs (Literature survey)** : This paper highlights that denoising regularization techniques, which have been effective on explicit representation, struggle to reliably train NeRFs in various situations despite previous research [1,2]. While some baselines may perform better under specific hyperparameters and scenarios, they often fail to converge in other cases and produce artifacts that look real data but are not part of the training set.
>
> As shown in Table 1, numerical results reveal that K-Planes exhibit greater scene-to-scene variance compared to the proposed method. Qualitatively, as demonstrated in Fig 5 and Figure E.12, K-Planes exhibit color-related issues due to denoising regularization. Particularly, as the denoising regularization value increases, color distortion becomes more pronounced. TensoRF experiences instability problems when dealing with sparse inputs, as seen in Table 3 and Table E.6, failing to converge when $\lambda_1$ exceeds 0.01. Stability and consistent performance across different scenarios are crucial, especially in few-shot training settings, where achieving the best performance in a few cases may not generalize well to other experiments.
>
> On the other hand, the proposed method consistently achieves top performance and robustly trains NeRFs, making it a reliable choice for various experimental cases. In few-shot scenarios, the significance of stable training is widely recognized, even if the proposed techniques are not entirely fresh. While we identify that classical approaches such as early-stopping and fewer capacity models are ineffective in this setting, they cannot eliminate all artifacts, it outperforms other baselines in both static and dynamic NeRF scenarios, indicating its stability and applicability in few-shot regimes. We have updated the literature survey to contain these arguments.
>
> [1] Fridovich-Keil, Sara, et al. "K-planes: Explicit radiance fields in space, time, and appearance." Proceedings of the IEEE/CVF Conference on Computer Vision and Pattern Recognition. 2023.
>
> [2] Cao, Ang, and Justin Johnson. "Hexplane: A fast representation for dynamic scenes." Proceedings of the IEEE/CVF Conference on Computer Vision and Pattern Recognition. 2023.

---

> ### Author Response · Authors · 2023-11-21
> **Response to r4Vs's review (2/3)**
>
> **The number of parameters and training time** : The proposed method does take longer compared to fast training methods like iNGP and TensoRF. However, it's worth noting that our approach still maintains reasonable training times. As shown in Table H.10, we've demonstrated that early stopping effectively preserves performance across all baselines, including our own, and all training procedures are completed in around 30 minutes.
>
> We conduct a comparison between TensoRF, K-Planes, and our model on the static NeRF dataset, limiting the training steps to 0.15 million and 8 views used for training. Rendering time is evaluated using 200 frames from the test dataset. We also experiment with less parameterized models. The numbers in brackets indicate the channel count in the multi-plane features each axis. We note that TensoRF(20) encountered training instability issues and failed to train on the scenes {chair, ficus, mic}, denoted by a hyphen. In the K-Planes model, featuring multi-resolution multi-plane characteristics, the total number of channel dimensions is the product of the number of resolutions and the channel dimension at each resolution.
>
> |   Model  Name   | # Params [M] | Avg.  PSNR | Avg. Training  Time [min] | Avg. Rendering  Time [min] |
> |:---------------:|:------------:|:----------:|:-------------------------:|:--------------------------:|
> | K_Planes (3*16) |      17M     |    23.95   |           17.61           |            6.83            |
> | K_Planes (2*16) |     4.4M     |    23.16   |           13.72           |            6.51            |
> | TensoRF (64)    |     17.3M    |    25.23   |            7.72           |            7.82            |
> | TensoRF (20)    |     6.1M     |      -     |             -             |              -             |
> | Ours (48)       |     6.0M     |    24.36   |           31.16           |            46.02           |
> | Ours (24)       |     3.0M     |    23.74   |           24.06           |            40.76           |
>
> The following table shows the experiment result for dynamic NeRF, including the number of parameters, average training time and rendering time. We constrain the training steps to 0.15 million and 25 views used for training. Rendering time is evaluated using 20 frames from the test dataset.
>
> |   Model  Name   | # Params [M] | Avg.  PSNR | Avg. Training  Time [min] | Avg. Rendering  Time [min] |
> |:---------------:|:------------:|:----------:|:-------------------------:|:--------------------------:|
> | K_Planes (3*32) |      18.6M     |    23.85   |           18.93           |            0.83            |
> | K_Planes (3*4) |     1.9M     |    23.41   |           13.29           |            0.78            |
> | HexPlane (72)    |     9.7M    |    24.00   |            6.78           |            0.60           |
> | HexPlane (6)    |     0.8M     |    22.08       |           6.38            |            0.68             |
> | Ours (48)       |     3.4M     |    25.17   |           12.22           |            2.14           |
> | Ours (24)       |     1.0M     |    25.10   |           8.77           |            1.73           |
>
>   * We include the results of iNGP in the manuscript
>
> An intriguing point to note is that when we reduce the number of parameters in the baselines to match ours, such as K-Plane, and TensoRF, their performance suffers, and they sometimes fail to converge. However, when we halve the number of parameters, our model's reported performance only experiences a slight decrease while still outperforming the baselines, and it also achieves slightly faster training and rendering speeds.
>
> Regarding rendering time, our method may not be the most efficient. However, when combined with recent rendering techniques employed in iNGP, it can significantly accelerate rendering speeds. Nevertheless, we believe that this paper's primary focus is on robust training and criticizing the side-effects of existing regularization on explicit representation. Therefore, the rendering pipeline is not the main concern here, and this issue does not align with the core message of the paper.
>
> Regarding dynamic NeRFs, the training and rendering times do not exhibit a significant gap compared to static NeRFs. We will provide more details on this result shortly.

---

> ### Author Response · Authors · 2023-11-23
> **Response to r4Vs's review (3/3)**
>
> **Real-world dataset** : We evaluate the proposed method on the real-world `Tanks and Temples` dataset, where it was compared with the baseline TensoRF models, including a regularization ($\lambda_1 = 0.001$).
> We selected the 'Tanks and Temples' dataset because it contains scenes where the camera distribution has objects-wise, rather than scenes with forward-facing camera distribution. As stated in the literature survey in the manuscript, training on object-facing data is more challenging when dealing with sparse input situations.
>
> In this table, we present SSIM of test dataset produced by the proposed method and baselines. All methods are trained on 10% of training views.
>
> |             Method | Barn  | Caterpillar | Family | Truck |
> |------------------:|-------|-------------|--------|-------|
> | TensoRF | 0.804 | 0.860       | 0.771  | 0.830 |
> | TensoRF ($\lambda_1=0.001$)    | 0.825 | 0.867       | 0.788  | 0.841 |
> | Ours      | 0.839 | 0.887       | 0.897  | 0.877 |
>
> The proposed method consistently achieves higher SSIM scores across all scenes, indicating its superior capability in preserving the structural integrity and overall composition of scenes.
>
> While PSNR is a valuable metric for image quality, it can be biased due to the lack of mask information for white backgrounds and the inclusion of full-resolution ($1920\times1980$) in the `tanks and temples` dataset. We show the PSNR result as follows.
>
> |             Method | Barn   | Caterpillar | Family | Truck  |
> |------------------:|--------|-------------|--------|--------|
> | TensoRF | 21.936 | 21.775      | 14.953 | 20.345 |
> | TensoRF ($\lambda_1=0.001$)    | 23.021 | 22.232      | 15.222 | 21.033 |
> | Ours      | 22.733 | 21.780      | 16.431 | 21.000 |
>
> In the revised manuscript, we have included both quantitative and qualitative results while varying the number of training views.
> Please refer to the section I in the appendix of the revised manuscript.
>
> To sum up, the proposed method distinguishes itself from the baselines through its robust ability to preserve the global context of scenes, handle sparse input data effectively, and render images that are visually realistic. The experimental results consistently demonstrates how the proposed method enhances its potential for broader application in real-world scenarios, where input data is often sparse and incomplete.

---

### Author Response · Authors · 2023-11-21
**General Response**

Thank you all for helpful suggestions.

We thank all the reviewers for their thorough and helpful comments. If any of our responses to individual reviewers below is insufficient, please feel free to ask any additional questions.

---
### **List of concerns we addressed in the new manuscript and the supplementary file**

- The number of parameters and training time for static NeRFs and dynamic NeRFs → Appendix H (reviewer r4Vs, D6bb,  CJze)
- Revision of literature survey → Section 2 (reviewer r4Vs)
- Experiments on the real-world dataset → Appendix I (reviewer r4Vs)
- Include additional animations of rendered images → Updated supplementary files (reviewer YmuZ)
- Include experiment results of K-Planes
- Modification of results from TensoRF due to an artificial scaling of the TV loss in the official codes
  (https://github.com/apchenstu/TensoRF/blob/9370a87c88bf41b309da694833c81845cc960d50/models/tensoRF.py#L196)



### **Discussion on reviews**
- Contribution and Novelty (all reviewers)

### **Minors**
- Fix typos of Eq.3  (reviewer fqvy)
- Revise the caption of Fig 3.  (reviewer D6bb)

---

### Meta-Review · Area_Chair_oFny · 2023-12-10

**Metareview:**

(a) This paper presents a method for enhancing neural radiance fields (NeRFs) for both static and dynamic scenes, especially in sparse input scenarios. The authors propose refined tensorial radiance fields that integrate coordinate-based networks, known for their bias towards low-frequency signals, with multi-plane encoding to capture fine-grained details. The paper claims that this integration effectively captures global context and details, overcoming limitations of existing multi-plane methods.

(b) Strengths: The use of a combined approach to address specific limitations in capturing global context in sparse input scenarios is reasonable. The responses to reviewer concerns are thorough, addressing aspects like training times, model sizes, and real-world dataset evaluations.

(c) Weaknesses:
1. The primary concern is the limited novelty of the approach. The method largely combines existing techniques without introducing substantial innovations.

2. Increased training and rendering times compared to existing methods are significant drawbacks, especially in the context of efficiency being a critical factor in NeRF applications. The improved performance comes at the expense of increased computational resources.

3. The paper lacks in-depth insights or significant analytical contributions, which is a critical issue for paper without substantial innovations.

**Justification For Why Not Higher Score:**

1. The primary concern is the limited novelty of the approach. The method largely combines existing techniques without introducing substantial innovations.

2. Increased training and rendering times compared to existing methods are significant drawbacks, especially in the context of efficiency being a critical factor in NeRF applications. The improved performance comes at the expense of increased computational resources.

3. The paper lacks in-depth insights or significant analytical contributions, which is a critical issue for paper without substantial innovations.

**Justification For Why Not Lower Score:**

N/A

---

### Decision · Program_Chairs · 2024-01-16

Reject